# PARP2 promotes Break Induced Replication-mediated telomere fragility in response to replication stress

Daniela Muoio [1], Natalie Laspata[1,2], Rachel L. Dannenberg[3], Caroline Curry[2], Simone Darkoa-Larbi[2], Mark Hedglin [3], Shikhar Uttam [4] & Elise Fouquerel [1] ✉

PARP2 is a DNA-dependent ADP-ribosyl transferase (ARTs) enzyme with Poly(ADP-ribosyl)ation activity that is triggered by DNA breaks. It plays a role in the Base Excision Repair pathway, where it has overlapping functions with PARP1. However, additional roles for PARP2 have emerged in the response of cells to replication stress. In this study, we demonstrate that PARP2 promotes replication stress-induced telomere fragility and prevents telomere loss following chronic induction of oxidative DNA lesions and BLM helicase depletion. Telomere fragility results from the activity of the break-induced replication pathway (BIR). During this process, PARP2 promotes DNA end resection, strand invasion and BIR-dependent mitotic DNA synthesis by orchestrating POLD3 recruitment and activity. Our study has identified a role for PARP2 in the response to replication stress. This finding may lead to the development of therapeutic approaches that target DNA-dependent ART enzymes, particularly in cancer cells with high levels of replication stress.

PARP2 is a member of the ADP-ribosyl transferase (ARTs) enzyme superfamily. ART enzymes catalyze an $NAD^+$-dependent post-translational modification of proteins called ADP-ribosylation[1]. They covalently bind the ADP-ribose moiety of the $NAD^+$ onto aspartic acid, glutamic acid, arginine, lysine, and serine residues of acceptor proteins[2-4]. While most of the ART family members are only capable of catalyzing mono(ADP-ribosyl)ation (MARylation), PARP2 can also add additional ADP-ribose units to form long-branched polymers composed of up to 100 ADP-ribose units (Poly(ADP-ribosyl)ation or PARylation)[1]. The most characterized enzyme of the ART family is its founding member PARP1. Initially identified for its role in the recognition of DNA strand breaks, including single- and double-strand breaks (SSBs and DSBs), and the orchestration of their repair[5,6], its activity is now known to be also triggered by a wide variety of DNA substrates such as DNA crosslinks, stalled replication forks, G-quadruplexes, and R-loops[7-12]. PARP1 is therefore involved in the response

to replication stress and is central to many DNA damage response (DDR) pathways, including SSB repair, base excision repair (BER), homologous recombination (HR), and alternative end joining (alt-EJ)[13]. The known functions of PARP2 in the DDR are largely redundant with those of PARP1. Similar to PARP1, depletion of PARP2 triggers sensitivity to ionizing radiation, alkylating agents, X-rays, and the radiomimetic drug bleomycin, suggesting that PARP2 plays an essential role in suppressing genomic instability in response to DNA damage by promoting SSB repair, DSB repair, and replication fork restart[14-18]. Additionally, similar to PARP1, PARP2 interacts with XRCC1, Polymerase beta, and Ligase III, critical components of the SSB and BER pathways[6], and can efficiently recruit XRCC1 and PNKP onto oxidized chromatin despite its weaker PARylation activity[19,20].

Distinct roles for PARP2 in DNA repair have recently emerged. Like PARP1, PARP2 stimulates DNA end resection for DSB repair[18]. However, unlike PARP1, which stimulates DNA end resection in the absence of

[1]UPMC Hillman Cancer Center, University of Pittsburgh Cancer Institute, Department of Pharmacology and Chemical Biology, Pittsburgh, PA 15213, USA. [2]Department of Biochemistry and Molecular Biology, Thomas Jefferson University, 233S. 10th street, Philadelphia, PA 19107, USA. [3]Department of Chemistry, The Pennsylvania State University, University park, State College, PA 16802, USA. [4]Department of Computational and Systems Biology, UPMC Hillman Cancer Center, University of Pittsburgh, 5117 Centre Avenue, Pittsburgh, PA 15213, USA. ✉e-mail: Elf115@pitt.edu

c-NHEJ proteins KU70/80 through the PARylation-dependent recruitment of the MRN complex[21,22], a direct interaction between PARP2 and the MRN complex has not been observed. Instead, PARP2's role seems to prevent 53BP1 binding to the breaks, thereby allowing CtIP/MRN complex recruitment. Notably, this function of PARP2 is independent of its ADP-ribosylation activity[18]. PARP2 also plays a critical role in preventing genomic instability associated with replication stress in rapidly proliferating cell types[23–27]. Accordingly, loss of PARP2 triggers replication stress in erythroblasts, as illustrated by an increase in phosphorylation of histone H2AX and single-strand DNA binding protein RPA in S-phase cells, as well as a rise in micronuclei causing defects in erythropoiesis in mice[23]. Additionally, PARP2 limits c-Myc-driven B-cell lymphoma expansion by preventing c-Myc-mediated replication stress and accumulation of DNA damage[26]. Finally, PARP2's activity is primarily activated by DNA nicks and gaps with 5′-phosphorylated ends, flaps and recombination intermediates[8,28–31]. Together, these studies suggest that PARP2 plays a specific role in ensuring genomic stability, particularly during replication. However, the molecular mechanisms involved remain elusive.

Replication stress is defined as a change in replication fork velocity caused by DNA damage, insufficient nucleotide availability, conflicts with other cellular processes, and oncogene overexpression. This leads to asymmetric fork progression, incomplete replication, fork collapse, and the generation of DNA lesions that ultimately cause genomic instability and contribute to the development of various diseases including cancer[32,33]. Several genomic regions are more prone to replication stress than others, including common fragile sites (CFS) and telomeres. Telomeres protect linear chromosomes by forming a t-loop with the assistance of a specific protein complex named shelterin. Due to their G-rich repetitive sequences, telomeres are inherently vulnerable to replication stress, as they can form stable G-quadruplex structures (G4s) that hinder replication fork progression[34,35]. Telomeric TTAGGG repeats are also highly sensitive to oxidative stress. Guanines are indeed readily oxidized due to their low redox potential leading to the formation of the adduct 8-oxoGuanine (8-oxoG)[36,37]. Thus, 8-oxoG lesions are formed more abundantly at telomeres than in other genomic loci upon exposure to oxidizing agents[38–41]. Multiple studies in human tissues, mice, and cell cultures have shown that chronic inflammation and oxidative stress are linked to accelerated telomere shortening and dysfunction. Accordingly, we have previously demonstrated that local chronic induction of 8-oxoG at telomeres in cancer cells leads to telomere shortening and instability[42]. Additionally, recent findings indicate that acute induction of telomeric 8-oxoG induces hallmarks of cellular senescence in non-diseased epithelial and fibroblast cells[43]. Both studies reported that the underlying mechanism was due to replication stress elicited directly by the 8-oxoG lesions[42,43].

An indication of replication stress at telomeres is the presence of fragile telomeres which can be detected in metaphase as multiple fluorescence in situ hybridization (FISH) foci on chromatid ends. Like in the case of CFS, telomere fragility is thought to result from replication stress because it was observed following replication inhibition by aphidicolin[44,45]. Specific to telomeres is a rise of fragile telomeres following the depletion of the Shelterin protein TRF1[44,45]. TRF1 facilitates telomere replication by recruiting the G4 unwinding helicase BLM[44,46]. Accordingly, the loss of BLM leads to increased fragility on the telomeric G-rich lagging strand[46,47]. A coping mechanism engaged upon aphidicolin-induced replication stress is the cleavage of the stalled replication forks by endonucleases followed by mitotic DNA synthesis (MiDAS)[47–50]. MiDAS has been described at both CFS and telomeres[47–50]. MiDAS is a form of Break-Induced Replication (BIR) that has been described to occur at telomeres upon DSB and oxidative stress induction[42,47,51]. This process requires DNA end resection by the MRN complex and synthesis by polymerase delta (Pol d)[52,53]. Interestingly, BIR competes with PARP1-dependent alt-EJ for the repair of telomeric DSBs occurring during replication stress and was shown to be responsible for the telomere fragility phenotype[47]. Given the redundant roles of PARP1 and PARP2 in the DDR mechanisms and the emerging specific roles of PARP2 in the replication stress response, we asked whether PARP2 also functioned in the replication stress response at telomeres.

In this study, we investigate the roles of PARP2 in the orchestration of the replication stress response at telomeres using a chemoptogenetic tool that induces oxidative DNA lesions[42] as well as following the depletion of the helicase BLM. We find that telomere fragility is promoted by PARP2 following replication stress induction. We demonstrate that PARP2 orchestrates the BIR pathway by promoting end resection and MiDAS. While end resection does not require its ADP-ribosylation activity, we show that it is required for MiDAS and that PARP2 can ADP-ribosylate the Polδ subunit POLD3 in vitro and in cells. Our findings point towards a role for PARP2 in the repair of collapsed replication forks during replication stress and highlight the significance of developing PARP-specific inhibitors for effective cancer treatment.

## Results

### Both PARP1 and PARP2 contribute to oxidative lesion repair at telomeres

To define the roles of PARP2 in the response to replication stress at telomeres, we leveraged the Fluorogen-Activated Peptide (FAP) tool, which allows for the targeted induction of 8-oxoG lesions in HeLa cells[42]. FAP is a chemoptogenetic system based on the affinity of the FAP peptide for the photosensitizer di-iodinated malachite green (MG2i) dye which excitation following exposure of the cells to a 660 nm wavelength light, generates singlet oxygen radical ($^1O_2$). When fused with the telomere binding protein TRF1, the FAP tool triggers the oxidation of guanines specifically at telomeres, resulting in the formation of 8-oxoG[42,54] (Fig. 1a). We have previously detected PARylation activity at telomeres following acute dye and light treatment[42]. To evaluate the contribution of each PARP enzyme in the early response to oxidative DNA lesion induction, we assessed their respective recruitment at telomeres using the Proximity ligation assay (PLA). Cells were incubated with the MG2i dye for 15 min and exposed to the red light for 5 min prior to fixation and staining with an antibody targeting the telomere binding protein TRF2 in combination with a PARP1 antibody or a PARP2 antibody. The ubiquitous oxidizing agent $H_2O_2$ was used as a positive control. The quantification of the number of foci formed thanks to the proximity of PARP1 or PARP2 with TRF2 revealed that both enzymes are efficiently recruited to telomeres immediately after acute dye and light exposure, and $H_2O_2$ treatment (Fig. 1b-d). It is noteworthy that the average number of PARP1/TRF2 PLA foci was lower after dye and light than after $H_2O_2$-treatment. Conversely, PARP2/TRF2 foci increased significantly in dye and light-treated cells compared to $H_2O_2$-treated cells. Our data align with previous studies that demonstrated the affinity of PARP1 for a broad variety of DNA ends and abasic sites (AP sites), including those induced by $H_2O_2$, while PARP2 selectively binds to 5′ phosphorylated ends, such as the ones that can form after 8-oxoG removal[8]. Consistent with this, inhibition of APE1 endonuclease (inhibitor III) that is responsible for generating 3′OH and 5′P ends after 8-oxoG removal by OGG1, significantly impacted the recruitment of PARP2 but not that of PARP1 (Supplementary Fig. 1a, b).

Next, we evaluated the role of each enzyme in preserving telomere integrity during acute oxidative stress. We depleted PARP1 and/or PARP2 in the HeLaFAP cell line using CRISPR. Two single-cell clones were expanded for each knock out (KO): PARP1KO (clones 6.10 and 3.4), PARP2KO (clones 22 and 28) and PARP1/2KO (clones 15 and 33, both generated using PARP1KO clone 6.10). Protein depletion and reduction of DNA-damage dependent ADP-ribosylation were confirmed through western blot analysis (Fig. 1e, Supplementary Fig. 1c).

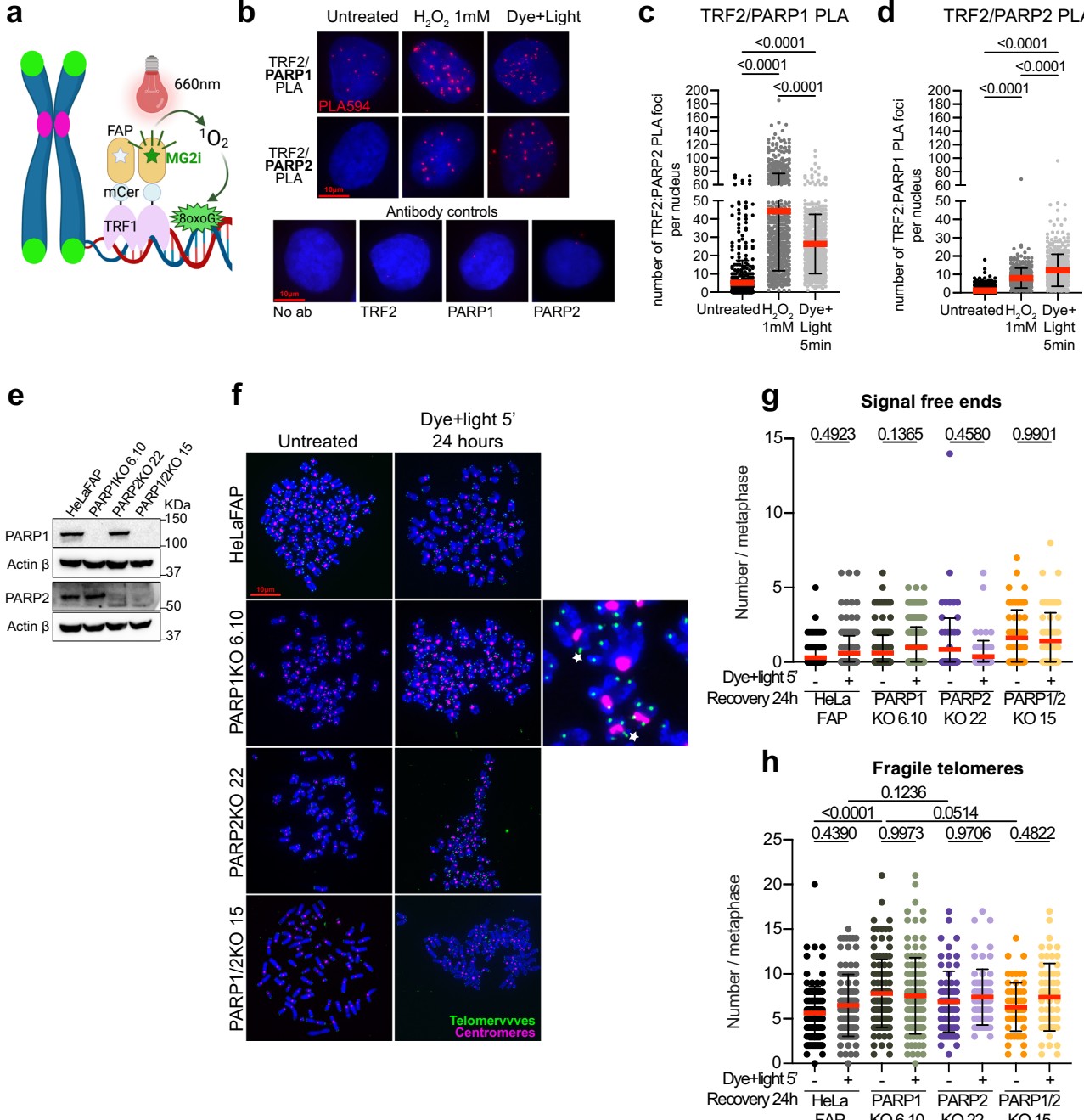

**Fig. 1 | Both PARP1 and PARP2 contribute to oxidative lesion repair at telomeres. a** Schematic of chemoptogenic tool used to induce telomeric 8oxoG (created with BioRender.com). **b** Representative images of TRF2:PARP1 and TRF2:PARP2 PLA foci (red) detected in HeLaFAP cells. Negative controls for the PLA are shown. **c**, **d** Quantification of the number of PLA TRF2:PARP1 (**c**) and TRF2:PARP2 (**d**) foci (red) per nucleus detected in HeLaFAP cells upon 1 mM $H_2O_2$ or after dye and light treatment. Each dot on the graph corresponds to a specific analyzed nucleus. At least 150 cells were analyzed per experiment. Red bars represent the mean ± SD from the indicated n number of nuclei analyzed from three independent experiments. *P*-values were obtained using ordinary one-way ANOVA. **e** Immunoblot of PARP1 and PARP2 in parental HeLaFAP cells and the KO clones. Actin was used as a loading control. **f** Representative images of telomere

FISH on metaphase chromosomes 24 h after acute dye and light exposure (5 min treatment). Fragile telomeres (white stars) are indicated. **g** Quantification of telomeric signal-free ends 24 h after acute dye and light exposure. Each dot represents a metaphase. At least 20 to 30 metaphases were analyzed per experiment. Red bars represent mean ± SD from *n* metaphases analyzed from three independent experiments. *P* values were obtained using ordinary one-way ANOVA. **h** Quantification of fragile telomeres 24 h after acute dye and light exposure. Each dot represents a metaphase. At least 20 to 30 metaphases were analyzed per experiment. Red bars represent mean ± SD from n metaphases analyzed from three independent experiments. *P*-values were obtained using ordinary one-way ANOVA. Source data are provided as a Source Data file.

Additionally, we confirmed a decrease in ADP-ribosylation activity at telomeres upon dye and light treatment using immuno-fluorescence (IF) coupled with telomere fluorescence in situ hybridization (telo-FISH) (Supplementary Fig. 1d, e). The depletion of PARP1 resulted in a decrease of telomeric PAR signal to a background level. Although ADP-

ribosylation activity was less affected in PARP2KO cells, there was a significant reduction in the PAR signal in these cells (Supplementary Fig. 1d, e). Moreover, we noted a slower 8-oxoG repair rate in PARP1KO cells compared to PARP2KO cells, as illustrated by a faster migration of telomeres on denaturing southern blot (Supplementary Fig. 1f, g).

Together, these data confirm a more significant contribution of PARP1 in the immediate response to oxidative DNA lesion induction.

Finally, teloFISH was performed on metaphase chromosomes to evaluate the impact of PARP1 and PARP2 depletion on telomere integrity. The induction of acute oxidative stress did not increase the number of telomere losses (Fig. 1f, g) or fragile telomeres (Fig. 1f, h) in cells lacking PARP1 or PARP2 compared to untreated cells. However, we noted a high basal level of fragile telomeres in PARP1KO cells, most likely arising from a basal level of replication stress in these cells (Fig. 1h). This is in line with the recently reported role of PARP1 in ensuring proper telomere replication by recruiting helicases BLM and WRN to unwind telomeric G-quadruplex structures[55]. In contrast, the PARP2KO cells did not display a significant number of fragile telomeres. Very interestingly, PARP1/2KO harbored fewer fragile telomeres than PARP1KO cells (Fig. 1h), indicating that deleting PARP2 from PARP1KO cells rescues this phenotype. This data suggests a role for PARP2 in the response to replication stress distinct from the one of PARP1.

## PARP2 depletion prevents replication stress mediated telomere fragility

To further elucidate the role of PARP2 in the replication stress response, we induced replication stress at telomeres using two approaches. Chronic exposure to oxidative stress has been shown to trigger replication stress at telomeres[42]. Thus, in our first approach, we asked whether repeated induction of 8-oxoG could exacerbate the telomere dysfunction phenotype. HeLaFAP, PARP1KO, PARP2KO, and PARP1/2KO cells were exposed to dye and light once per day for 24 days, as previously described (Fig. 2a)[42]. We followed the cell growth rate of each clone during the experiment and examined telomere dysfunctions after the 18th dye and light treatment (Fig. 2a). The slopes of the cell population doubling curves indicated that chronic oxidative stress affects cell growth, which is exacerbated following PARP1 depletion but not PARP2 (Fig. 2b and Supplementary Fig. 2a). Interestingly, we observed that the depletion of both enzymes has a significantly greater impact on cell growth than upon their individual depletion. This suggests a synergistic role for PARP1 and PARP2 in the response to replication stress via distinct pathways. Likewise, the analysis of mean telomere length (MTL) by southern blot showed a progressive telomere shortening that was the most pronounced in PARP1/2KO cells (Supplementary Fig. 2b, c). Importantly, further examination of individual telomeres by telo-FISH on mitotic chromosomes indicated a marked rise of telomere losses in these cells but not in HeLaFAP or PARP1KO and PARP2KO cells (Fig. 2c, d). Strikingly, chronic oxidative stress also led to a significant increase in the number of fragile telomeres in PARP1KO cells but not in PARP2KO or PARP1/2KO cells (Fig. 2c, e). These findings suggest that PARP2 depletion in cells lacking PARP1 can counteract the occurrence of fragile telomeres. The data were replicated with additional clones harboring different MTLs (Supplementary Fig. 2d–h) indicating that the sensitivity of telomeres to PARP1 and PARP2 loss is independent of telomere length (Supplementary Fig. 2d–h).

Telomere fragility was first reported as a result of the loss of the telomere binding protein TRF1[44]. TRF1 facilitates telomere replication by recruiting crucial factors such as the Bloom (BLM) helicase to unwind the G-quadruplex structures that can form on the G-rich lagging strand during replication[44,46]. Consequently, loss of TRF1 or BLM helicase triggers replication stress and leads to telomere fragility[44,47]. Thus, in a second approach, we induced replication stress by depleting BLM using siRNA in our PARP1 and/or PARP2-deficient clones. Reduction of BLM expression was verified by western blot 48 h after transfection (Fig. 2f). As expected, BLM knockdown increased the number of fragile telomeres in HeLaFAP cells which was further exacerbated in PARP1KO cells (Fig. 2g and Supplementary Fig. 2i). Strikingly, BLM depletion-mediated telomere fragility was prevented in both cell lines lacking PARP2, whereas PARP1/2KO cells harbored a

slight increase in the number of telomere losses (Fig. 2g, Supplementary Fig. 2i, j). Collectively, our data suggest that PARP2 is responsible for the occurrence of telomere fragility in PARP1-depleted cells subjected to replication stress.

## Telomere fragility is mediated by PARP2 and its catalytic activity during replication stress

To confirm our hypothesis of the direct involvement of PARP2 in the mechanisms driving telomere fragility, we exogenously re-expressed PARP1 or PARP2 as C- or N-terminal fusions to a 3x Flag tag in PARP1/2KO cells (Fig. 3a, b) (FLAG-PARP1 and PARP2-FLAG). Telo-FISH staining on metaphase spreads 48 h after BLM siRNA transfection revealed that both PARP1 and PARP2 complementation promote a decrease in the number of telomere losses in PARP1/2KO cells upon BLM depletion (Fig. 3d). Importantly, PARP1 re-expression did not have any impact on the number of fragile telomeres in PARP1/2KO cells. In contrast, PARP2 re-expression led to an increase in the number of fragile telomeres (Fig. 3e). Moreover, re-expression of a catalytically dead mutant of PARP2 obtained by mutating the catalytic glutamate 558 into alanine (E558A) (Fig. 3b, c and Supplementary Fig. 3a, b), failed to restore telomere fragility in BLM-depleted cells (Fig. 3e).

We also induced replication stress by triggering chronic oxidative stress using our FAP tool. We first noted that the growth defect of PARP1/2KO cells, initially observed upon repeated dye and light exposure, was rescued by re-expression of PARP2-FLAG, but not of the E558A mutant (compare Supplementary Fig. 2a with Supplementary Fig. 3c, d). Importantly, we observed that PARP2-FLAG expression in PARP1/2KO cells restored telomere fragility while rescuing telomere loss in cells undergoing replication stress (Fig. 3f, g). However, the E558A mutant-expressing cells still harbored a high level of telomere losses and a basal level of fragile telomeres (Fig. 3f, g). The lack of impact of the E558A was not due to an alteration of its recruitment to telomeres. This was confirmed by PLA assay using anti-TRF2 and anti-Flag antibodies which revealed efficient recruitment of both FLAG-PARP2 and E558A mutant to telomeres upon induction of oxidative stress (Supplementary Fig. 3e, f). Finally, restoration of the telomere fragility phenotype and prevention of telomere loss were also observed in additional PARP1/2KO clones (PARP2-FLAG, clone 11 and E558A, clone 4), whose transgene expressions and ADP-ribosylation levels were closer to those of the endogenous PARP2 (Supplementary Fig. 3g–j). Collectively, these data demonstrate that PARP2, but not PARP1, can directly promote replication-stress mediated telomere fragility dependent on its activity.

## PARP2 orchestrates mitotic DNA synthesis at telomeres upon replication stress

Cells subjected to replication stress can display under-replicated DNA by the end of the interphase, which requires mitotic DNA synthesis (MiDAS)[47–50]. Therefore, we investigated whether PARP2 is involved in the MiDAS mechanisms at telomeres upon replication stress induced by chronic oxidative stress. Cells were exposed to 18 cycles of dye and light and arrested in the G2 phase of the cell cycle using a CDK1 inhibitor. Cells were then released for 1 h in media containing EdU and mitoses were collected by shake-off (Fig. 4a). Scoring of EdU-positive chromatid ends revealed an increase in conservative DNA synthesis dependent on replication stress (staining at a single chromatid; Fig. 4b) in cells lacking PARP1 but not in PARP2-depleted cells (Fig. 4c–g). An increase in semi-conservative DNA synthesis was also observed, as evidenced by staining at both sister chromatids (Fig. 4b) in PARP1/2KO cells after dye and light treatment, suggestive of HR events (Fig. 4c–g). Critically, the re-expression of PARP2-FLAG but not of the E558A mutant restored conservative DNA synthesis at telomeres (Fig. 4h, i). The same results were observed in our additional clones indicating that MiDAS occurs independent of the mean telomere length and the levels of PARP2 re-expression and activity (Supplementary Fig. 4a–e).

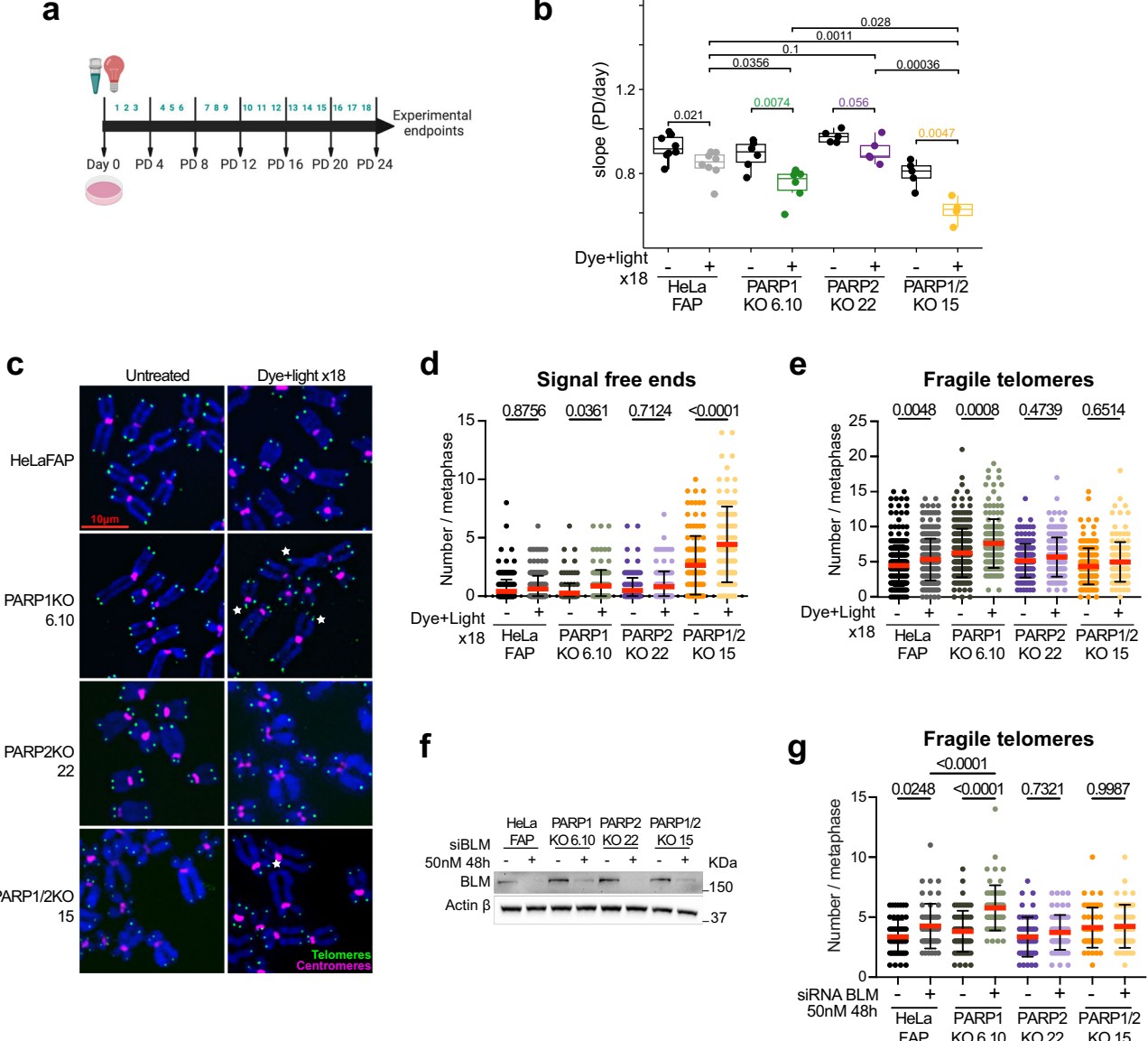

**Fig. 2 | PARP2 depletion prevents replication stress mediated telomere fragility. a** Schematic of the chronic induction of $_1O^2$. Arrows indicate the days when cells were passaged or harvested and not treated with dye and light (created with BioRender.com). **b** Mean projection slopes obtained from the population doubling curves of 4 to 8 independent experiments. *P*-values were obtained using two-sided Welch's two-sample t-test. Minima, maxima, bounds of box and whiskers and percentile are indicated in the Source Data file. **c** Representative images of telomere FISH on metaphase chromosomes 24 h after the 18th dye and light exposure. Telomeric signal-free ends and fragile telomeres (white stars) are indicated. **d** Quantification of telomeric signal-free ends 24 h after the last dye and light exposure treatment (N18). Each dot represents a metaphase. At least 30 metaphases were analyzed per experiment. Red bars represent mean ± SD from n metaphases analyzed from four independent experiments. *P*-values were obtained

using ordinary one-way ANOVA. **e** Quantification of fragile telomeres 24 h after the last dye and light exposure treatment (N18). Each dot represents a metaphase. At least 30 metaphases were analyzed per experiment. Red bars represent mean ± SD from n metaphases analyzed from four independent experiments. *P*-values were obtained using ordinary one-way ANOVA. **f** Western blot analysis of BLM in Hela-FAP, PARP1KO, PARP2KO, and PARP1/2KO cells treated with 50 nM BLM siRNA for 48 h. Actin was used as a loading control. **g** Quantification of fragile telomeres detected by FISH in HelaFAP, PARP1KO, PARP2KO, and PARP1/2KO after knockdown of BLM with siRNA. Each dot represents a metaphase. At least 20 to 30 metaphases were analyzed per experiment. Red bars represent mean ± SD from n metaphases analyzed from two independent experiments. *P*-values were obtained using ordinary one-way ANOVA. Source data are provided as a Source Data file.

Taken together, our data indicate that PARP2 and its catalytic activity orchestrate the conservative DNA synthesis at telomeres subjected to replication stress.

## PARP2 stimulates BIR-mediated DNA resection and strand invasion

MiDAS at one chromatid end is a feature of the break-induced replication pathway (BIR). Interestingly, BIR has been shown to be responsible for telomere fragility upon replication stress following

BLM depletion[47]. Thus, we next asked whether fragile telomeres arising in PARP1-deficient cells upon chronic oxidative stress were also the result of BIR. To test this, we first used telomere chromosome orientation FISH (CO-FISH) staining. Because BIR involves DNA synthesis on both strands of one chromatid, CO-FISH, in which the staining process involves the selective removal of the newly synthesized DNA strands[56], is unable to detect the resulting fragile telomeres (Fig. 5a). We detected the parental G-rich lagging strand, originally carrying the 8-oxoG lesions, using the PNA probe TelC-Alexa488 (green) and the leading

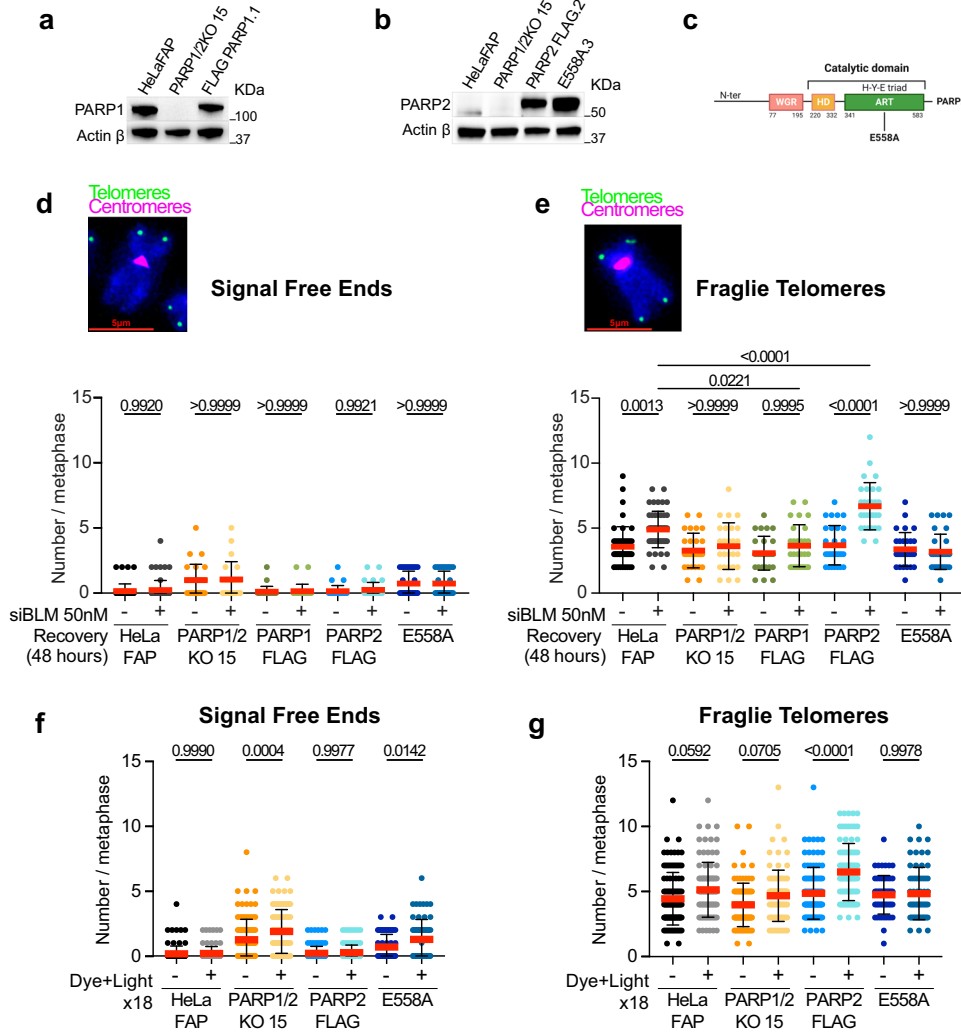

**Fig. 3 | Telomere fragility is mediated by PARP2 and its catalytic activity during replication stress. a** PARP1/2KO cells were transfected with the pCMV-PARP1-3xFlag-WT plasmid, and protein expression was analyzed by immunoblotting using an anti-PARP1 antibody. Actin was used as a loading control. **b** PARP1/2KO cells were transfected with the pLVX-IRES-puro-PARP2-FLAG plasmid, and its version containing the point mutations, and protein expression was analyzed by immunoblotting using an anti-PARP2 antibody. Actin was used as a loading control. **c** Modular structures of the ART enzymes PARP2. The amino acid numbers indicate domain boundaries. The dark line indicates the point mutation E558A of the catalytic glutamate of the ART domain. **d** Quantification of telomeric signal-free ends detected by FISH in HelaFAP, PARP1/2KO, PARP1-FLAG, PARP2-FLAG, and E558A cell lines after knockdown of BLM with siRNA. Each dot represents a metaphase. Red bars represent mean ± SD from n (35 to 65) metaphases analyzed. *P*-values were obtained with ordinary one-way ANOVA. **e** Quantification of fragile telomeres detected by FISH in HelaFAP, PARP1/2KO, PARP1-FLAG, PARP2-FLAG, and E558A cell lines after knockdown of BLM with siRNA. Each dot represents a metaphase. Red bars represent mean ± SD from n (35 to 65) metaphases analyzed. *P*-values were obtained with ordinary one-way ANOVA. **f** Quantification of telomeric signal-free ends detected by FISH in HelaFAP, PARP1/2KO, PARP2-FLAG, and E558A cell lines 24 h after the last dye and light exposure (N18). Each dot represents a metaphase. At least 20 metaphases were analyzed per experiment. Red bars represent mean ± SD from n metaphases analyzed from three independent experiments. *P*-values were obtained with ordinary one-way ANOVA. *P*-values were obtained using ordinary one-way ANOVA. **g** Quantification of fragile telomeres detected by FISH in HelaFAP, PARP1/2KO, PARP2-FLAG, and E558A cell lines 24 h after the last dye and light exposure (N18). Each dot represents a metaphase. At least 20 metaphases were analyzed per experiment. Red bars represent mean ± SD from n metaphases analyzed from three independent experiments. *P* values were obtained using ordinary one-way ANOVA. Source data are provided as a Source Data file.

C-rich strand, using the PNA probe TelG-Cy3 (red) and compared the number of fragile telomeres obtained with the one observed upon regular FISH staining using the TelC-Alexa-488 probe only. Cells underwent 18 cycles of dye and light treatment, and telomeres on metaphase spreads were stained following the FISH or the CO-FISH protocols. FISH staining confirmed an increase in fragile telomeres exclusively in PARP1KO cells (Fig. 5b and Supplementary fig. 5a–c). However, CO-FISH staining prevented the detection of fragile telomeres after dye and light treatments in these cells confirming that BIR is responsible for telomere fragility in cells lacking PARP1. The remaining fragile telomeres occurred mostly within the lagging strand and were indicative of basal replication stress. As expected, both FISH

and CO-FISH staining did not detect fragile telomeres in the other cell lines (Supplementary fig. 5a–c). FISH staining further confirmed that re-expressing PARP2-FLAG in PARP1/2KO cells but not the E558A mutant, restored telomere fragility that was therefore no longer detected following CO-FISH staining (Supplementary fig. 5c, d). These results were consistent across all additional clones (Supplementary fig. 5e–i). Overall, these data suggest that PARP2 directs BIR-dependent DNA synthesis.

BIR repairs one-ended DNA breaks by utilizing a homologous template. This process involves extensive 5′ to 3′ end resection by the Mre11/Rad50/Nbs1 (MRN) complex, which generates 3′ single-stranded DNA (ssDNA) stretches that are bound by RPA proteins. Subsequently,

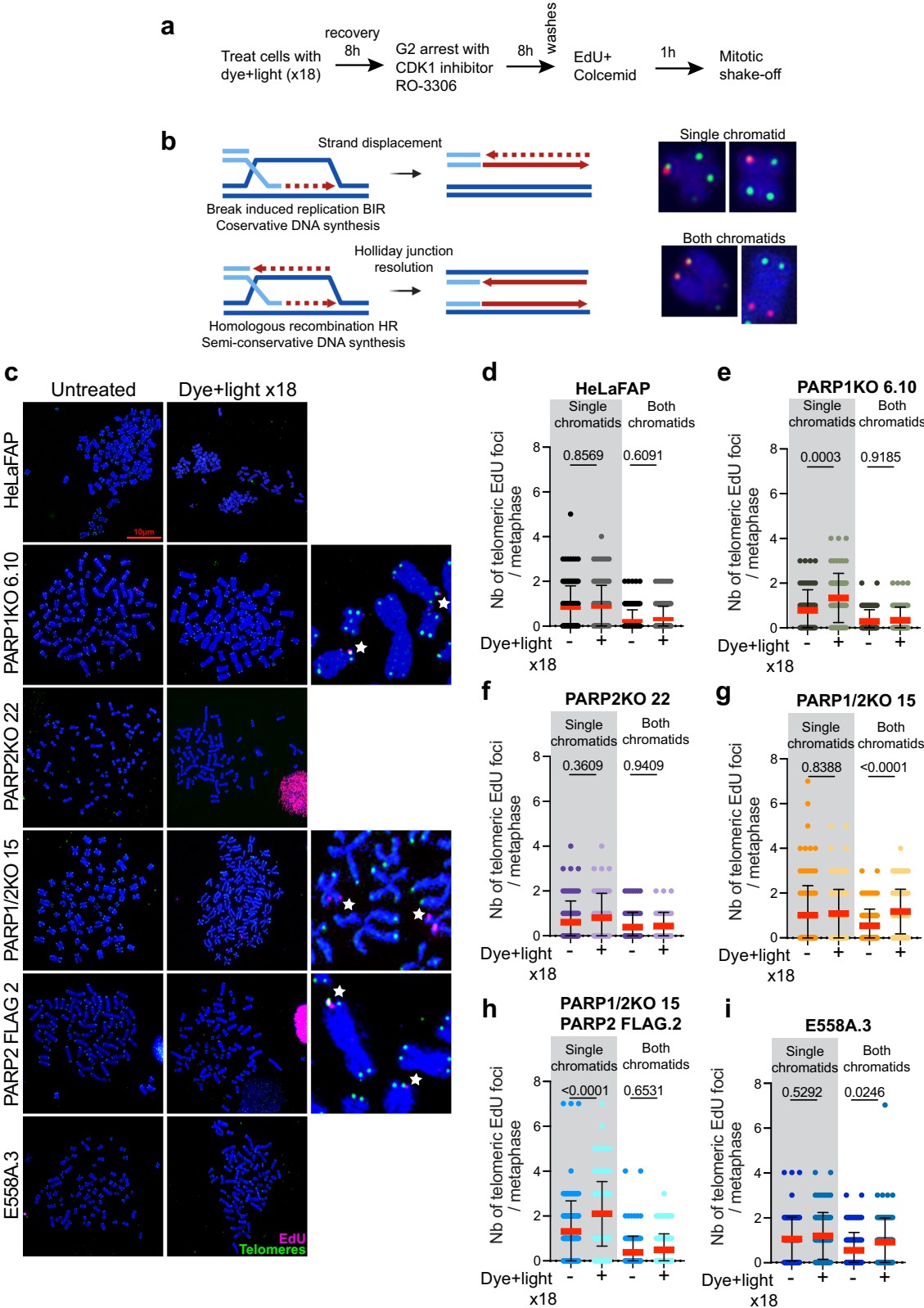

**Fig. 4 | PARP2 ORCHESTRATES MITOTIC DNA SYNTHESIS AT TELOMERES UPON REPLICATION STRESS. a** Schematic of the experimental steps conducted to measure MiDAS at telomeres. **b** Models illustrating conservative DNA synthesis during break-induced replication (top) and semi-conservative DNA synthesis during homologous recombination (bottom). Red, DNA synthesis (chromosomes show a corresponding MiDAS pattern). **c** Representative images of EdU incorporation (pink) at telomeres (green) on a single or both chromatid ends of metaphase chromosomes (blue) after 18 dye and light exposures. The last column shows the enlargement of both types of MiDAS (white starts). **d–i** Quantification of EdU incorporation at telomeres on single or both chromatid ends of metaphase chromosomes from every cell line after 18 dye and light treatments. Each dot represents a metaphase. At least 20–30 metaphases were analyzed per experiment. Red bars represent mean ± SD from n metaphases analyzed from three independent experiments. *P* values were obtained using ordinary one-way ANOVA. Source data are provided as a Source Data file.

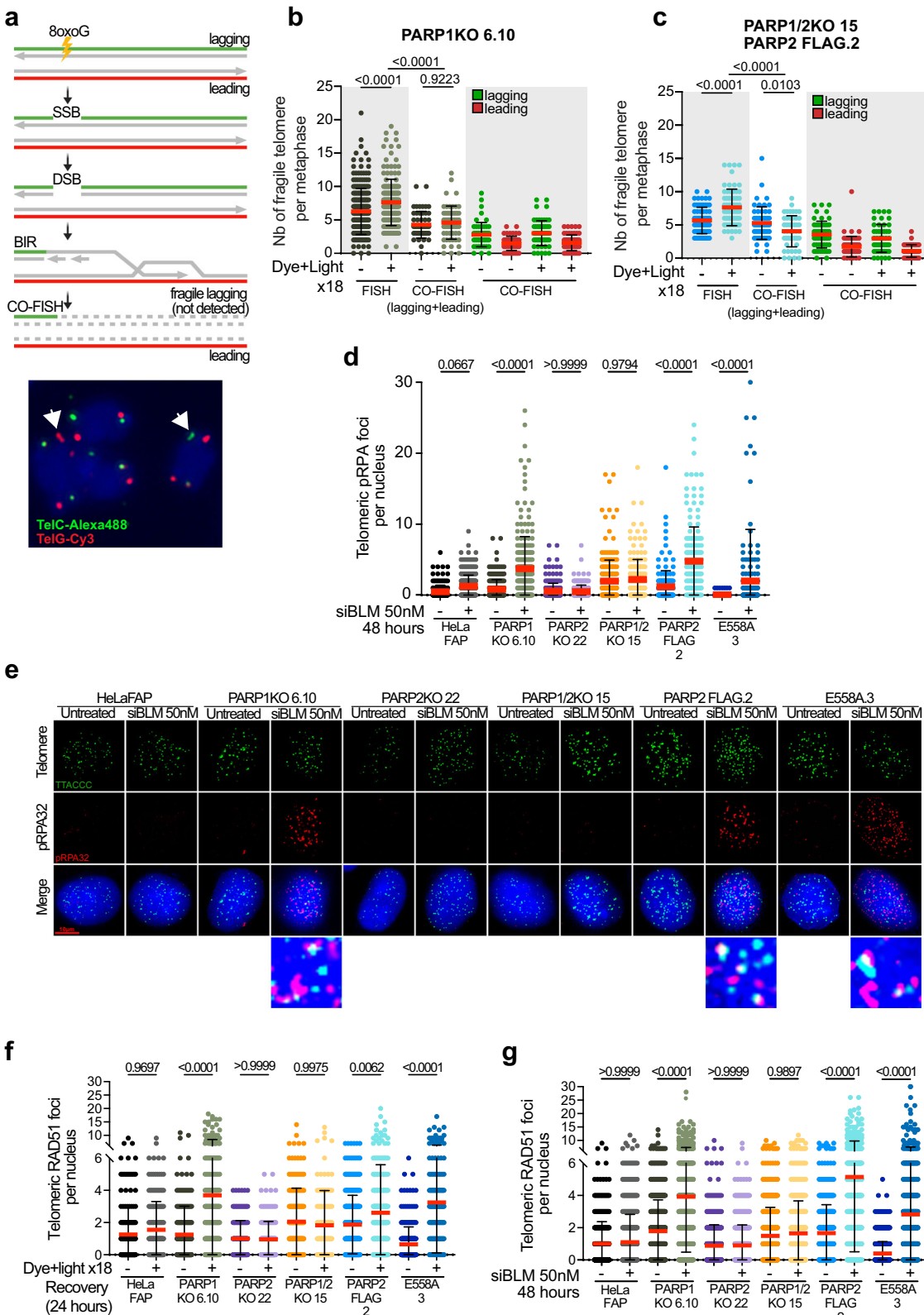

a Rad52-mediated formation of a Rad51 nucleofilament takes place, which searches for the homologous template. Inhibiting the Mre11 nuclease with Mirin after chronic dye and light treatment not only reduced the basal number of fragile telomeres in the HeLaFAP cell line but prevented their increase upon oxidative stress in PARP1KO cells (Supplementary fig. 6a). This confirms the occurrence of BIR-dependent DNA end resection. We next asked whether PARP2 and its PARylation activity played an active role in the end resection step of

BIR at telomeres following replication stress. To test this, cells were subjected to replication stress via chronic oxidative stress and/or BLM depletion (Supplementary fig. 6b), and RPA and Rad51 recruitment to telomeres was followed by IF and telo-FISH. BLM depletion resulted in an increase of RPA foci at telomeres in PARP1KO cells and in PARP1/2KO cells complemented with PARP2-FLAG (Fig. 5d, e). Similarly, BLM depletion and chronic oxidative stress led to the recruitment of RAD51 to telomeres in PARP1KO cells but not in the cell lines lacking PARP2

**Fig. 5 | PARP2 STIMULATES BIR-MEDIATED DNA END RESECTION AND STRAND INVASION. a** Model for BIR-mediated fragile telomere formation and their removal after CO-FISH. Representative image of telomere CO-FISH. The Leading strand DNA was hybridized with a red probe and the lagging strand was hybridized with a green probe. White arrows show the fragile telomeres. **b**, **c** Quantification of fragile telomeres in PARP1KO and PARP2-FLAG cells after 18 dye and light treatments detected by FISH compared to fragile telomeres detected by CO-FISH on samples derived from BrdU/BrdC-labeled cells, and quantification of leading- and lagging-end telomeres. Each dot represents a metaphase. At least 20 metaphases were analyzed per experiment. Red bars represent mean ± SD from n metaphases analyzed from three independent experiments. *P*-values were obtained using ordinary one-way ANOVA. **d** Quantification of the number of pRPA32 foci colocalizing with telomeres per nucleus. Cells were fixed 48 h after the knockdown of BLM with siRNA. Each dot on the graph corresponds to a specific analyzed nucleus. At least 100 cells were counted for each experiment. Red bars represent mean ± SD from n nuclei analyzed from three independent experiments. Statistical analysis was performed using ordinary one-way ANOVA. **e** Representative images of pRPA32 foci (in red) combined with FISH staining of telomeres (in green) in HeLaFAP, PARP1KO, PARP2KO, PARP1/2KO, PARP2-FLAG, and E558A cell lines. Cells were fixed 48 h after the knockdown of BLM with siRNA. The last row corresponds to zoomed-in squares showing marked pRPA32 foci colocalizing with telomeres. **f** Quantification of the number of RAD51 foci colocalizing with telomeres per nucleus. Cells were fixed 24 h after the last dye and light treatment (N18). Each dot on the graph corresponds to a specific analyzed nucleus. At least 200 cells were analyzed. Red bars represent mean ± SD. Statistical analysis was performed using ordinary one-way ANOVA. **g** Quantification of the number of RAD51 foci colocalizing with telomeres per nucleus. Cells were fixed 48 h after the knockdown of BLM with siRNA. Each dot on the graph corresponds to a specific analyzed nucleus. At least 100 cells were counted for each experiment. Red bars represent mean ± SD from n nuclei analyzed from three independent experiments. Statistical analysis was performed using ordinary one-way ANOVA. Source data are provided as a Source Data file.

(Fig. 5f, g, Supplementary fig. 6c–e). However, the re-expression of PARP2-FLAG in PARP1/2KO cells significantly increased Rad51 foci colocalizing to telomeres. Interestingly, the re-expression of the catalytic mutant E558A also restored Rad51 recruitment (Figs. 5f, g, S6c–e) and RPA foci formation at telomeres, albeit not at the level observed upon PARP2-FLAG complementation (Fig. 5d, e) suggesting that PARP2 enzymatic activity may partly control RPA foci formation. These data suggest that BIR-dependent DNA end resection and strand invasion necessitate PARP2 but not its catalytic activity.

## PARP2 is required for POLD3 recruitment to telomeres

Our data demonstrated that telomere fragility and MiDAS are dependent on PARP2 activity (Figs. 4 and 5). Since the end resection step of BIR did not seem to require ADP-ribosylation (Fig. 4), we hypothesized that PARP2 catalytic activity is necessary for the DNA synthesis step of BIR. MiDAS was shown to be dependent on the POLD3 subunit of polymerase delta which is recruited to telomeres upon BLM depletion to mediate BIR[47,48]. Moreover, a recent study has demonstrated that PARP1/2 catalytic activity was required for the assembly of POLD3 on stalled replication forks to promote BIR[57]. Therefore, we asked whether PolD3 recruitment to telomeres was dependent on PARP2 specifically. Replication stress was induced by chronic oxidative stress or BLM depletion and POLD3 recruitment to telomeres was assessed using PLA with POLD3 and TRF2 antibodies. Consistent with a role of PARP2 in POLD3 recruitment, chronic oxidative stress caused a significant increase in PLA foci in PARP1KO cells, which was prevented in PARP1/2KO cells (Fig. 6a, b Supplementary fig. 7a, b). Conversely, cells complemented with PARP2-FLAG or the E558A mutant displayed an increase of foci upon treatment (Fig. 6a, b and Supplementary fig. 7b), suggesting that POLD3 recruitment to telomeres depends on PARP2 but does not require ADP-ribosylation activity. We obtained similar results upon BLM depletion except that, very interestingly, POLD3 recruitment seemed largely enhanced in cells expressing PARP2 catalytic mutant (Fig. 6c, d) suggesting retention of the polymerase subunit when PARP2 is unable to be activated. We next tested the impact of inhibiting the Poly-(ADP-ribose) degrading enzyme PARG on POLD3 recruitment to telomeres. We treated BLM-depleted PARP1/2KO cells complemented with FLAG-PARP2 with PARG inhibitor PDD00017273 (PARGi) for 24 h and performed PLA assays as previously described. PARG inhibition efficiency was assessed by western blot using anti-PAR 10H antibody (Fig. 6e). We confirmed that BLM depletion triggered an increase of PLA foci formation between POLD3 and TRF2 antibodies (Fig. 6f, g). However, PARGi treatment prevented the formation of PLA foci suggesting that excessive poly-ADP ribosylation activity impacts the occupancy of POLD3 at the site of replication stress. Collectively, these data indicate that PolD3 recruitment at telomeres subjected to replication stress, is promoted by PARP2, and does not require its catalytic activity. They also demonstrate that a balance between the ADP-ribosylation synthesis and degradation is required for POLD3 function.

## POLD3 is a target of PARP2

Our data indicates that POLD3 recruitment to telomeres does not occur through the ADP-ribose. Thus, we next tested whether PARP2 and POLD3 could interact directly. We used PLA assay using a mouse anti-flag antibody to detect PARP2-FLAG and the E558A mutant. Because the PolD3 antibody used in our previous PLA experiment was raised in mice, we overexpressed POLD3 fused with RFP and used a rabbit anti-RFP antibody to detect POLD3. POLD3-RFP expression in PARP1/2KO cells, complemented by PARP2-FLAG or the E558A mutant, was confirmed by western blot using POLD3 antibody (Fig. 7a). The PLA assay revealed an interaction between wild-type PARP2-FLAG and POLD3 in BLM-depleted cells (Fig. 7b, c and Supplementary fig. 8a). Interestingly, cells expressing the PARP2 catalytic mutant displayed a high basal level of PLA foci in BLM-proficient cells that was increased upon siRNA treatment (Fig. 7b, c). Conversely, treatment of PARP2-FLAG expressing cells with PARGi led to a reduction of PLA foci (Fig. 7d, e). Because BLM can resolve G-quadruplexes not only at telomeres, we also verified that our data were reproducible at these genomic loci specifically. To do this, we quantified the number of PLA foci that colocalized with the mCerulean protein of our telomeric FAP tool expressed in the cells treated with siRNA BLM (Fig. 1a). As expected, we observed an increase of PLA foci after BLM depletion in both cell lines that was enhanced in cells expressing the PARP2 catalytic mutant E558A (Supplementary fig. 8b, c). Finally, the interaction between POLD3 and PARP2 was also confirmed by immunoprecipitation (Supplementary fig. 8d, e). These data reveal an interaction between PARP2 and POLD3, which stability increases when PARP2 is unable to perform its enzymatic activity.

High-resolution mass spectrometry and proteome-wide analysis identified several ADP-ribosylation sites in POLD3 amino acid sequence[58]. A recent study has also reported that PolD3 is a target for PARP1/ PARP2 and that its site-specific ADP-ribosylation is required for BIR activity, replication fork recovery, and genome stability[57]. Based on these findings, we hypothesize that PARP2 may ADP-ribosylate PolD3 at collapsed forks. To test this hypothesis, we performed an in vitro PARP2 hetero-modification assay using the recombinant Polymerase Delta (Polδ) protein complex comprising the subunits p125 (POLD1), p66 (POLD3) and p50 (POLD2) (Supplementary fig. 8f). Strong PARP2 auto-ADP-ribosylation activity was detected upon incubation of recombinant GST-hPARP2 with its co-factor NAD[+] and DNaseI-activated DNA (activated DNA) but not in the absence of NAD[+] or activated DNA (Fig. 7f). PARP2 auto-ADP-ribosylation was also detected upon addition of Polδ. Consistent with ADP-ribosylation of POLD3, the poly/mono-ADP-ribose antibody detected a band at the p66 molecular weight that was accompanied by a smear. Interestingly,

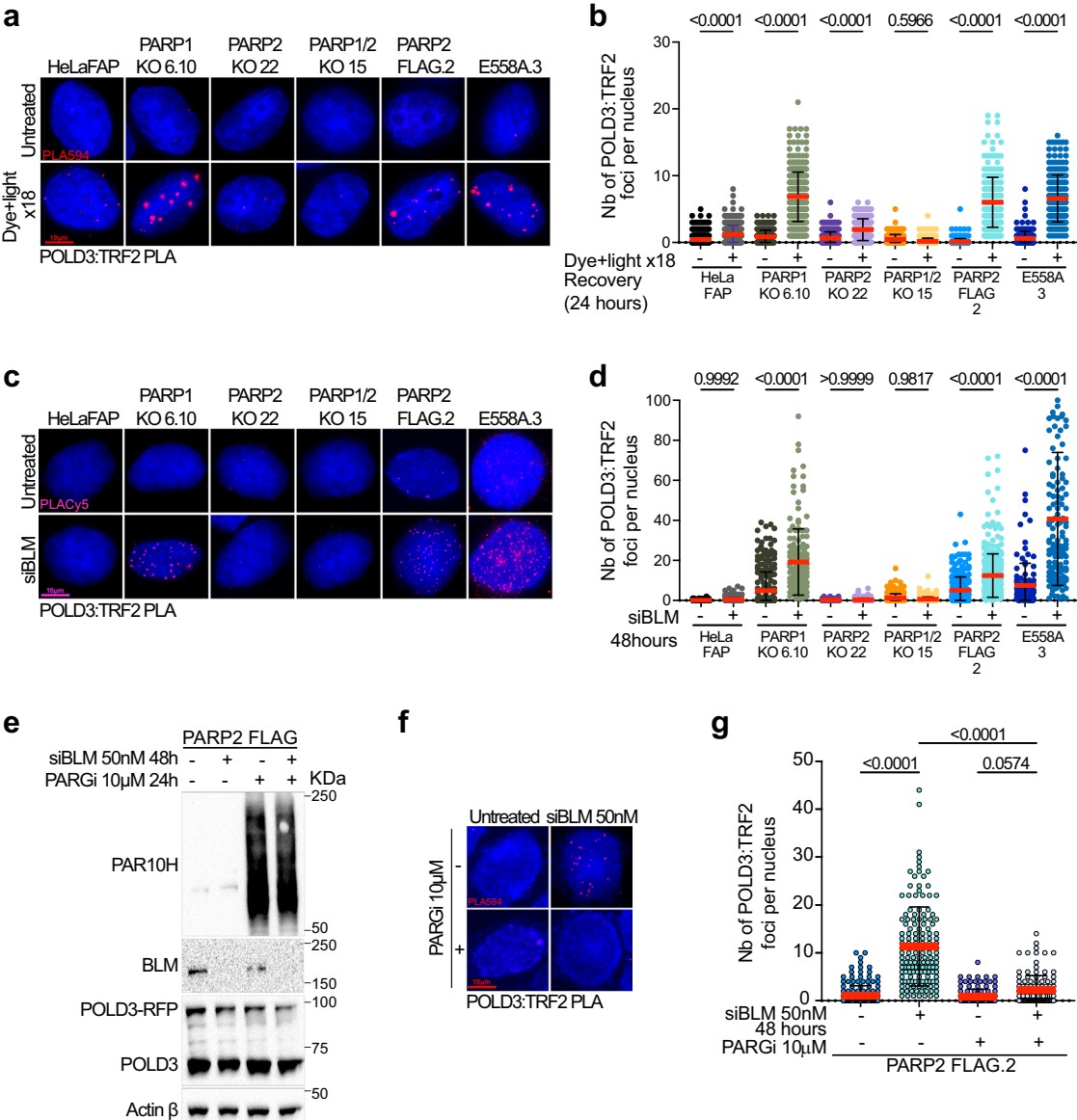

**Fig. 6 | PARP2 IS REQUIRED FOR POLD3 RECRUITMENT TO TELOMERES.**
**a** Representative images of POLD3:TRF2 PLA foci (red) detected in HeLaFAP, PARP1KO, PARP2KO, PARP1/2ko, PARP2-FLAG, and E558A cells after chronic induction of oxidative stress using dye and light. **b** Quantification of the number of POLD3:TRF2 PLA foci (red) per nucleus detected in HeLaFAP, PARP1KO, PARP2KO, PARP1/2ko, PARP2-FLAG and E558A cells. Cells were fixed 24 h after the last dye and light treatment (N18). Each dot on the graph corresponds to a specific analyzed nucleus. At least 300 cells were analyzed. Red bars represent mean ± SD. *P*-values were obtained using ordinary one-way ANOVA. **c** Representative images of POLD3:TRF2 PLA foci (pink) detected in HeLaFAP, PARP1KO, PARP2KO, PARP1/2ko, PARP2-FLAG, and E558A cells after depletion of BLM with siRNA. **d** Quantification of the number of POLD3:TRF2 PLA foci (pink) per nucleus detected in HeLaFAP, PARP1KO, PARP2KO, PARP1/2ko, PARP2-FLAG and E558A cells. Cells were fixed 48 h after the knockdown of BLM with siRNA. Each dot on the graph corresponds to a

specific analyzed nucleus. At least 300 cells were analyzed. Red bars represent mean ± SD. *P*-values were obtained using ordinary one-way ANOVA. **e** Immunoblot showing accumulation of PAR from PARP2-FLAG cell extracts treated with PARGi 10 mM for 24 h. Cells were depleted for BLM with 50 nM BLM siRNA for 48 h and were transfected with the POLD3-RFP plasmid. Protein expression was analyzed using anti-BLM and anti-POLD3 antibodies. Actin was used as a loading control. **f** Representative images of POLD3:TRF2 PLA foci (red) detected in PARP2-FLAG cells after depletion of BLM with siRNA, and inhibition of PARG. **g** Quantification of the number of POLD3:TRF2 PLA foci (red) per nucleus detected in PARP2-FLAG cells after knockdown of BLM with siRNA and PARG inhibition. Each dot on the graph corresponds to a specific analyzed nucleus. At least 100 cells were counted for each experiment. Red bars represent mean ± SD from n nuclei analyzed from two independent experiments. *P*-values were obtained using ordinary one-way ANOVA. Source data are provided as a Source Data file.

additional signals were also detected at the p125 and p50 molecular weights, suggesting that the three Polδ subunits tested can be ADP-ribosylated. To confirm POLD3 ADP-ribosylation in cells, we next conducted a PLA assay using the ADP-ribose and POLD3 antibodies in all our cell lines treated with BLM siRNA (Fig. 7g, h and Supplementary fig. 8g). We observed the formation of PLA foci in all our cell lines after treatment except in PARP1/2KO and in cells re-expressing the PARP2

E558A mutant (Fig. 7g, h and Supplementary fig. 8g). Interestingly, the number of PLA foci in PARP2Ko cells was similar to the number of foci in HeLaFAP cells and significantly lower than the number of foci quantified in PARP1KO cells. While these data indicate that both PARP1 and PARP2 seem capable of targeting POLD3, they also suggest that under replication stress conditions, PARP2 contributes to POLD3 targeting more significantly than PARP1.

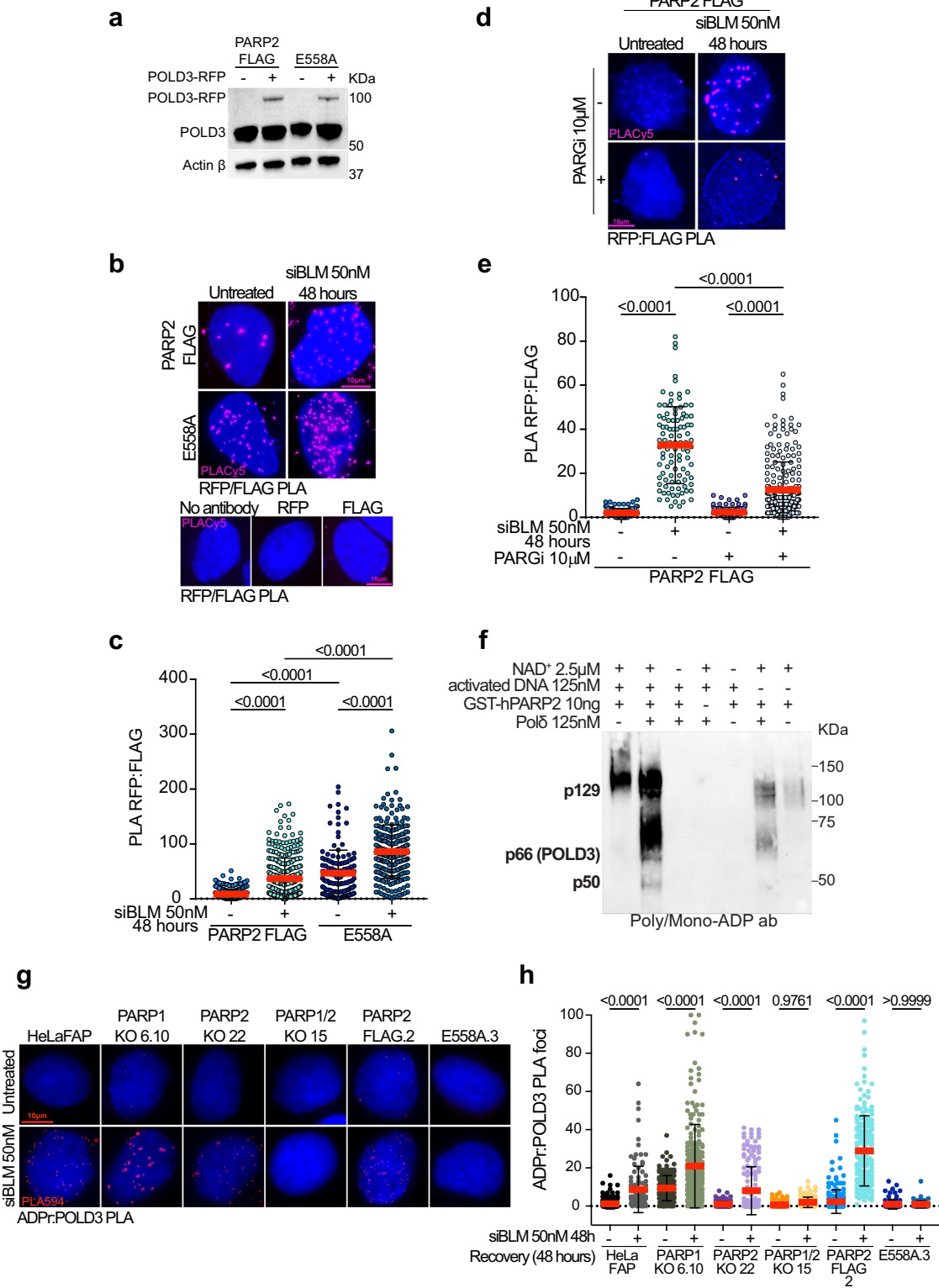

## Discussion

In this study, we uncover functions for PARP2 in the cellular response to replication stress. We notably demonstrate that PARP2 promotes telomere fragility at telomeres by orchestrating the BIR pathway, which repairs collapsed forks induced by chronic oxidative stress or BLM depletion. Our data also suggest that PARP2 contributes to preventing telomere loss by promoting telomere fragility (Fig. 8).

To induce replication stress, we induced chronic oxidative stress or depleted the G-quadruplex unwinding helicase BLM. In both

approaches, we observed that BIR-dependent telomere fragility was significantly increased in cells lacking PARP1. This can be explained by the prominent contribution of the more abundant and active PARP1 in repairing oxidative lesions, and its role in resolving G-quadruplexes at telomeres during replication[55], thereby preventing replication stress. Consistent with this, cell growth was impacted only in cells lacking PARP1 during chronic exposure to oxidative stress. At telomeres, Slx4 and Slx1, a scaffold protein and an endonuclease respectively, perform cleavage of stalled forks, generating DSBs[47,59,60] and initiating BIR[47].

**Fig. 7 | POLD3 IS A TARGET OF PARP2. a** PARP2-FLAG and E558A cells were transfected with the POLD3-RFP plasmid, and protein expression was analyzed by immunoblotting using a POLD3 antibody. Actin was used as a loading control. **b** Representative images of RFP:FLAG PLA foci (pink), and antibody controls, detected in PARP2-FLAG and E558A cells. **c** Quantification of the number of RFP:FLAG foci (pink) per nucleus detected in PARP2-FLAG, and E558A cells after knockdown of BLM with siRNA and overexpression of POLD3. Each dot on the graph corresponds to a specific analyzed nucleus. At least 100 to 300 cells were counted per condition. Red bars represent mean ± SD from n nuclei analyzed from two independent experiments. *P*-values were obtained using ordinary one-way ANOVA. **d** Representative images of RFP:FLAG PLA foci (pink) detected in PARP2-FLAG cells after depletion of BLM with siRNA, and inhibition of PARG. **e** Quantification of the number of RFP:FLAG PLA foci (pink) per nucleus detected in PARP2-FLAG cells after knockdown of BLM with siRNA, and PARG inhibition. Each dot on the graph corresponds to a specific analyzed nucleus. At least 100 to 300

cells were counted per condition. Red bars represent mean ± SD from n nuclei analyzed from two independent experiments. *P*-values were obtained using ordinary one-way ANOVA. **f** Heteromodification of POLD3 subunit of human DNA polymerase d by PARP2. Purified Pold was incubated with hPARP2 and activity buffer containing +/− $NAD^+$ and activated DNA. PARylation levels were analyzed by immunoblot with an anti-Poly/mono-ADP ribose antibody. **g** Representative images of ADPr:POLD3 PLA foci (red) detected in HeLaFAP, PARP1KO, PARP2KO, PARP1/2ko, PARP2-FLAG, and E558A cells after depletion of BLM with siRNA. **h** Quantification of the number of ADPr:POLD3 PLA foci (red) per nucleus detected in HeLaFAP, PARP1KO, PARP2KO, PARP1/2ko, PARP2-FLAG and E558A cells. Cells were fixed 48 h after the knockdown of BLM with siRNA. Each dot on the graph corresponds to a specific analyzed nucleus. At least 100 to 300 cells were analyzed per condition. Red bars represent mean ± SD. *P*-values were obtained using ordinary one-way ANOVA. Source data are provided as a Source Data file.

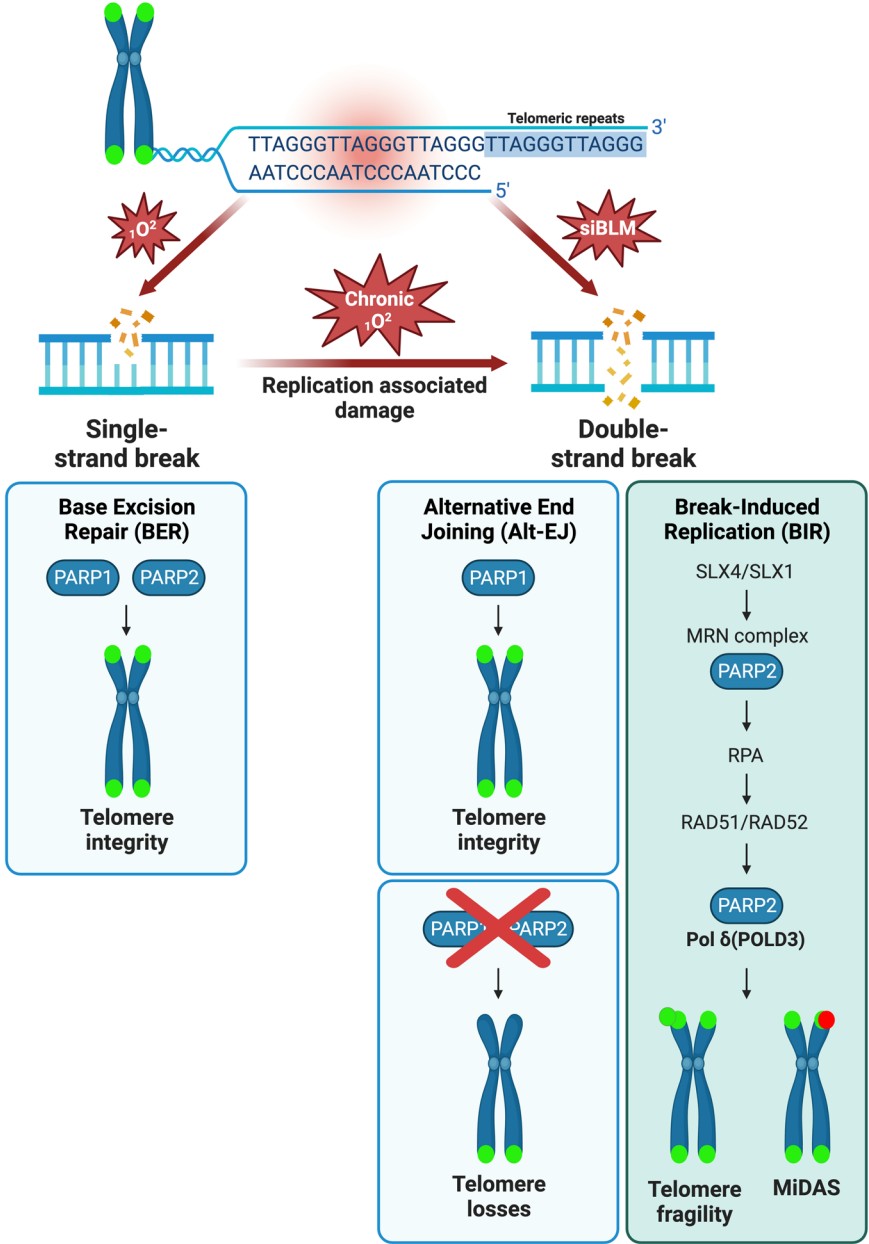

**Fig. 8 | Working model.** During replication stress, PARP1 promotes Alt-EJ to preserve telomere integrity while PARP2 orchestrates the BIR pathway during which it promotes DNA end resection and mitotic DNA synthesis by regulating PolD3 recruitment and activity. PARP2-dependent BIR triggers telomere fragility. Absence of PARP1 and PARP2 triggers telomere loss (created with BioRender.com).

PARP1-dependent Alt-EJ is a pathway engaged for the repair of internal telomeric DSBs[61], and it was found to compete with BIR for the repair of telomeric DSBs upon BLM loss[47], demonstrated by an increase of fragile telomeres upon PARP1 inhibition or depletion of Ligase 3[47]. Consistent with this, we report that the fragile phenotype induced by BLM depletion is not restored by re-expression of PARP1 in PARP1/2KO cells, unlike PARP2.

Interestingly, we highlighted that PARP2 is involved in two different crucial steps of the BIR pathway that are DNA end resection and DNA synthesis. PARP2 exhibits a strong preference for DNA break ends that harbor a 5′-phosphate[8]. Additionally, Slx1, a GIY-YIG endonuclease, cleaves replication forks and generates nick products with 5-P ends[62,63], making it an ideal substrate for PARP2 binding to initiate BIR at telomeres. Strikingly, we demonstrated that while DNA end resection does not require PARP2 enzymatic activity, it is required for the DNA synthesis step since MiDAS is abrogated in PARP1/2KO cells expressing PARP2 catalytic mutant. In agreement with our data, a previous study reported that PARP2 could promote end resection of genomic DSBs independently of PAR synthesis, by limiting 53BP1 to access the DSB ends, thereby stimulating homologous repair[18,61].

Surprisingly, our different PLA assays investigating the BIR polymerase POLD3 show that PARP2 enzymatic activity is not required for its recruitment to telomeres or its interaction with PARP2. In fact, POLD3 is more strongly recruited to telomeres in the cells expressing PARP2 catalytic mutant, and we also observed a stronger interaction with the PARP2 catalytic mutant. Nevertheless, we also demonstrate that PARP2 can ADP-ribosylate POLD3 in vitro and in cells, in line with a recent study[57]. POLD3 is one of the accessory subunits of the human DNA polymerase delta (Pol d) which also comprises two other accessory subunits (POLD2 and POLD4) and a large catalytic subunit (POLD1) that carries DNA polymerase and exonuclease activity[64]. POLD2 and POLD3 are also subunits of Pol z, a translesion synthesis (TLS) polymerase, whose other subunits are REV3 and REV7[65,66]. Pol z, along with the scaffolding factor REV1, can operate in G2 to promote replication of UV-damaged DNA[67]. Based on these observations, a very recent study has demonstrated that both Pol z and Pol d are required to promote MiDAS in a sequential manner[68]. More specifically, Pol z seems to act upstream of Pol d, which is recruited through an interaction of the REV1 interacting region (RIR) of POLD3 with REV1 and via a polymerase switch mechanism that remains unclear. Interestingly, POLD3 RIR comprises 2 residues that were found to be PARylated[58]. It is possible that ADP-ribosylation of RIR allows for the disruption of POLD3's interaction with PARP2 and its subsequent recruitment by REV1 which would enable the polymerase switch for the completion of DNA synthesis. Consistent with this hypothesis we report that POLD3 recruitment and interaction with wild-type PARP2 is disrupted by PARG inhibition. Future studies are warranted to determine whether Pol z and/or other TLS polymerases are specifically involved in MiDAS at telomeres. For instance, DNA polymerase eta (Pol η) was found to coordinate with Pol d to ensure recombination-associated DNA synthesis at ALT telomeres[69]. ALT or alternative lengthening of telomeres is a telomere maintenance mechanism engaged in about 10-15% of cancers in lieu of the most used telomerase-dependent process. ALT relies on BIR[51] and several groups have shown that ALT cells harbor significantly higher levels of spontaneous telomeric BIR and MiDAS events than telomerase positive cancer cells[49,51,70]. They also display a high level of fragile telomeres. However, BIR is active in non-ALT cells harboring DBSs (this work and[42,47]) and it is possible that different TLS polymerases are required that would depend on the cell type, telomere maintenance mechanism and protein environment which differs at ALT telomeres.

PAR metabolism has been demonstrated to be crucial in driving key steps of ALT mechanism. In line with our data obtained with PARGi, treatment of ALT cells with the same inhibitor triggers a reduction of BIR-induced DNA synthesis and diminishes the recruitment of POLD3 at telomeres while PARPi treatment slightly increases BIR-induced DNA synthesis and does not alter POLD3 localization to telomeres[71]. These data support our model in which PAR metabolism is responsible for the regulation of POLD3 localization and activity at telomeres. Together, they highlight the significance of maintaining a tight balance between PAR synthesis and degradation for an efficient BIR process.

Finally, PARP2 but not PARP1, was found to localize to ALT-associated promyelocytic leukemia bodies (APBs), which are the DNA synthesis centers of ALT cells[72]. This study did not report the impact of PARP2 loss on telomeres in U2OS cells. However, it was found that PARP2-deficient mouse primary cells harbored increased levels of telomere signal free ends (fragile telomeres were not scored before 2009) as well as the heterogeneity of telomere length, a feature of ALT cells[72]. Finally, our findings identified a specific function of PARP2 at telomeres in conditions of replication stress that corroborates an already reported unique role of this enzyme in mitigating genomic replication in cancer cells undergoing replication stress-driven oncogene dysregulation[26]. Collectively, these studies may encourage the development of therapeutic approaches differentially targeting PARP1 and PARP2, especially in ALT cancers and more generally in cancer cells harboring high levels of replication stress.

## Methods
### Cell culture and cell line generation
HeLaFAP cells were generated previously[42]. They were cultured at 5% oxygen in Dulbecco's Modified Eagle Medium (DMEM) containing 4 g/l glucose (GIBCO) and supplemented with 10% Fetal Bovine Serum (GIBCO), 1x penicillin/streptomycin (Life Technologies) and 500 µg/ml G418. PARP1KO cells were obtained by infection of the HeLaFAP cells with lentivirus containing a pLentiCRISPR v2 plasmid expressing both *S. pyogenes* Cas9 and guide RNAs targeting exon 2 for clone 6.10 (gRNA 6: GCTTCTGGAAGGTGGGCCAC) and exon 4 for clone 3.4 (gRNA 3: ATTGACCGCTGGTACCATCC) of PARP1 gene (Genscript). Lentiviral particles were produced in Hek293T cells (ATCC) using the GeCKO system protocol. Briefly, approximately $2.2 \times 10^5$/ml Hek239 cells were seeded in antibiotic-free growth media (DMEM + 10% iFBS) in a 6-well plate. The next day, for each well, 3 mL of TransIT-LT1 (MirusBio) was added to a 12 mL of OptiMEM (GIBCO) media and mixed with 500 ng of pLentiCRISPR v2 vector was co-transfected with 50 ng of the packaging plasmid pVSVg (Addgene 8454) and 500 ng of envelope plasmid psPAX2 (Addgene 12260) contained in 37.5 mL OptiMEM. After incubation at room temperature for 30 min, transfection mixes were added dropwise to the Hek293T cells at 80% confluency and incubated for 18 h at 37 °C. The next day, transfection media was replaced with 2 ml of High-BSA 293 T growth media (DMEM + 10% iFBS + 1 g/100 mL BSA + 1x Pen/Strep). The first virus harvest was performed 24 h after recovery and HeLaFAP was infected by incubation with the filtrated Hek293T conditioning media supplemented with 10 mg/ml of polybrene. This procedure was repeated twice. After overnight incubation with infected media, HeLaFAP cells were left to recover for 8 h before selection with 1.5 mg/ml puromycin and single-cell cloning. Each expanded clone was tested for PARP1 expression by western blot and one clone generated from each guide RNA was selected to conduct experiments.

*Parp2* gene knockout in HeLaFAP and PARP1KO cells was achieved by direct transfection of CRISPR Cas9 RNP complexes. Multiguide RNA mixture containing 3 different guide RNAs targeting exon 2 of *Parp2* gene (#1: G*A*A*AGCAAAAGAGUUAAUAA, #2: C*U*G*GCAUCUACGA-GUUUUCU, #3: A*G*G*ACAGAAGACAAGCAAGA) and *S. pyogenes* Cas9 nuclease 2NLS enzyme were purchased from Synthego. CRISPR Cas9 RNP was transfected using Lipofectamine CRISPRMAX Cas9 transfection reagent (Invitrogen) and following manufacturer recommendations. Briefly, 50000 cells were seeded in a 24-well plate. The next day, Cas9 RNPs were assembled in 25 ml of OptiMEM by mixing 1250 ng of *S. pyogenes* Cas9 nuclease 2NLS enzyme with 240 ng of guide RNAs and 2.5 ml of Cas9 Plus reagent. In a second tube, 1.5 ml of CRISPRMAX

reagent was diluted in 25 ml of OptiMEM. The mixture containing the assembled RNPs was added into the tube containing the CRISPRMAX reagent and incubated for 10 min at room temperature before being added to the cells. Once confluent, transfected cells were passaged in a 6-well plate followed by single-cell cloning. Each expanded clone was tested for PARP2 expression by western blot.

PARP1, PARP2, and PARP2 point mutant E558A complemented cells were obtained after infection of PARP1/2KO cells with lentiviral particles containing either pCMV-PARP1-3xFlag-WT or pLVX-IRES-puro-PARP2 vector. The pLVX-IRES-puro-PARP2 E558A proteins were obtained by mutation of the pLVX-IRES-puro-PARP2 plasmid using the QuikChange II XL Site-Directed Mutagenesis Kit (Agilent) and verified by sequencing. Viral particle generation, infection, and clone selection were performed as described above. All cell lines were cultured at 5% $O_2$ in DMEM containing 4 g/l of glucose, complemented with 10% Fetal Bovine Serum, 1x penicillin/streptomycin, and appropriate selection antibiotic.

For siRNA knockdown experiment, Dharmacon ON-TARGETplus SMARTpool siRNA for human BLM was purchased from Horizon (#L-007287-00-0005). For BLM knockdown, HeLaFAP cells were seeded at 70% confluency in standard growth media. Transfection was performed by mixing 50 nM of BLM siRNA in serum-free DMEM with DharmaFECT 1 transfection reagent (Horizon #T-2001-02) for 30 min at RT, during which the cell media was replaced with antibiotic-free DMEM + 10% FBS. The reagents were added to the cells and incubated overnight. The next day, the media was replaced with standard growth media, and the cells were allowed to recover for 48 h. Knockdown efficiency was validated by Western blot on a standard SDS-PAGE gel using anti-BLM (kind gift from R. O'Sullivan lab) and anti-b-actin (Sigma #088M4804) antibodies.

For POLD3 overexpression, HeLaFAP cells complemented with FLAG-PARP2 or E558A mutant were seeded in normal growth media, so they were 70-80% confluent at the time of transfection. The POLD3 p-TAG-RFP plasmid was used to transfect cells with Lipofectamine 3000 in OptiMEM. Cells were incubated overnight. The next day, the media was replaced with standard growth media, and the cells were allowed to recover over the day. POLD3 overexpression was validated by western blot using an anti-POLD3 (Abnova #H00010714-M01) antibody.

Cell lines were authenticated previously (O'Sullivan et al. 2014; Fouquerel et al. 2019). For derivatives of these cells generated in this study, the genotyping was verified by PCR amplification and sequencing of the relevant loci and protein levels established by western blotting with appropriate antibodies. All these cell lines were tested for mycoplasma contamination on a monthly basis using LookOut kit to confirm that they test negative for mycoplasma infection.

### Cell treatment for 8-oxoG induction
Acute and chronic singlet oxygen production was performed as previously described in ref. 42. The MG2i dye was used at 100 nM final concentration in OptiMEM. The cells were exposed to a 660 nm LED light with a radiant flux density of 0.14 W/cm² for 5 min (unless indicated otherwise) delivering a total of 41.5 J/cm², to trigger excitation of the FAP-bound MG2I dye. For chronic induction of telomeric 8-oxoG, $3 \times 10^5$ cells were seeded in 10 cm dishes and treated as described above, every 24 h for 3 consecutive days. Every 4th day, cells were harvested to perform analyses, and $3 \times 10^5$ cells were re-seeded to undergo a new cycle of exposure for another 3 days. Cells undergo a total of 18 exposures for 24 days.

### Telomeres restriction fragment analysis for mean telomere length
Telomere length analysis was performed as previously described in ref. 73 with minor modifications. Briefly, genomic DNA was isolated from cells using the QIAGEN Tip-100 according to the manufacturer's

instructions. After resuspension in TE, 3 μg of genomic DNA were digested with a cocktail of 4 restriction enzymes (AluI, HphI, MnlI, and HinfI 0.5 U each, NEB) overnight at 37 °C. Telomere restriction fragments for length analysis were resolved by pulse field gel electrophoresis on a 1% Certified Megabase Agarose gel (Biorad) in 0.5X TBE. Samples were electrophoresed at 14 °C and 6 V with a 1 s initial switch, and 6 s final switch for 12–15 h using a CHEF-DR II apparatus (BioRad). The gel was dried under vacuum at 50 °C for 2 h and stained with SYBR green, before denaturation (0.5 M NaOH, 1.5 M NaCl) and neutralization (0.5 M Tris pH 8, 1.5 M NaCl). After incubation for 30 min in hybridization buffer containing 5x Denhardt's solution (0.1% Ficoll 400, 0.1 % Polyvinylpyrrolidone, 0.1 % BSA), gels were probed overnight at 42 °C with radio-labeled (CCCTAA)[4] probe. The gels were washed with 2 X SSC, 0.1 % SDS, 0.1 X SSC, and 2 X SCC 10 min each, before exposing on a phosphorimager screen and imaged with a Typhoon RGB phosphoimager (Phosphostorage). Telomere probe radio-labeling was performed using OptiKinase and 30 mCi of g32P-ATP (PerkinElmer). Mean telomere restriction fragment lengths (MTL) were calculated using ImageQuant and Telorun analysis.

### Denaturing southern blot
Southern blot in denaturing conditions was performed as described previously in[74] with slight modifications. Genomic DNA was isolated using Qiagen genomic-tips 100/G. Each buffer was supplemented with Butylated hydroxytoluene and Deferoxamine mesylate salt (100 mM final each) until loading of the cell extracts on the genomic tips. Genomic DNA purification was then performed following manufacturer recommendations. 3 mg of DNA was digested overnight at 37 °C with restriction enzymes HphI, AluI, HindIII, MnlI and fractionated on 0.8% agarose gels containing 50 mM NaOH and 1 mM EDTA. Electrophoresis was run at 150 V for 1 h followed by a 50 V run for 24 h in the cold room. Gels were bathed in neutralization buffer (0.5 M Tris-HCl at pH 7.5, 1.5 M NaCl) for 30 min at room temperature and then dried for 2 h at 50 °C. Dried gels were rehydrated for 15 min in MilliQ water, incubated at 42 °C in 5 X SSC containing 1x SYBR green for 30 min and imaged with a Typhoon RGB phosphoimager (Cy2). Next, gels were treated in denaturation buffer for 15 min (0.5 M NaOH, 1.5 M NaCl), rinsed in MilliQ water for 10 min, and neutralized for 15 min (0.5 M Tris-HCl at pH 7.5, 1.5 M NaCl). After incubation for 30 min in hybridization buffer containing 5x Denhardt's solution (0.1 % Ficoll 400, 0.1% Polyvinylpyrrolidone, 0.1% BSA), gels were probed overnight at 42 °C with radio-labeled (CCCTAA)[4] probe. The gels were washed with 2 X SSC, 0.1 % SDS, 0.1 X SSC, and 2 X SCC 10 min each, before exposing on a phosphorimager screen and imaged with a Typhoon RGB phosphoimager (Phosphostorage). Telomere probe radio-labeling was performed using OptiKinase and 30 mCi of g32P-ATP (PerkinElmer).

### Population doubling measurement
The population doubling (PD) values were calculated using the mathematical formula $PD = [(\ln(N2)) - (\ln(N1))] / \ln(2)]$. N1 is the initial number of cells plated and N2 is the final number of cells counted. The PD curves were obtained using the sum of the individual PDs calculated every 4 days.

### Western blotting
Cells were harvested by trypsinization and lysed in RIPA buffer (150 mM NaCl, 1% NP40, 0.5% DOC, 0.1% SDS, 50 mM Tris-HCl at pH 8.0) supplemented with PMSF 1 nM and with Roche protease inhibitor cocktail tablets 1X for 30 min on ice and then centrifuged at maximum speed in a microfuge for 15 min at 4 °C. Protein concentrations were determined with the Pierce™ BCA Protein Assay Kit (Thermo Fisher), and 10–30 mg of protein in 20–40 mL of Laemmli buffer (Bio-rad) was electrophoresed on 4–12% precast Bis-Tris gels (Thermo Fisher) before transferring to nitrocellulose membranes (GE). The membranes were

blocked in 5% milk in TBS-Tween (TBST) and incubated with primary antibodies in 5% milk overnight, followed by three washes with TBST. The membranes were then incubated with horseradish peroxidase-conjugated goat anti-mouse or donkey anti-rabbit secondary antibodies for 1 h, followed by three washes with TBST before imaging. For PAR10H immunoblot, cells were washed in PBS 1X and scraped from the surface of a 6-well plate into Laemmli sample buffer and boiled for 10 min at 95 °C prior to loading on a precast Bis-Tris gel.

Antibodies used were anti-Poly(ADP-Ribose)10H mouse monoclonal (Enzo Life Science #ALX-802-220-R100, WB dilution 1:1000), anti-PARP1 C-2-10 mouse monoclonal (Enzo Life Science #BML-SA249 BML-SA249, WB dilution 1:1000), anti-PARP-2 (4G8) mouse monoclonal (Enzo Life Science #ALX-804-639-L001 WB dilution 1:1000), anti-DYKDDDDK (FLAG) (FG4R) mouse monoclonal (Invitrogen #MA1-91878, WB dilution 1:1000), anti-POLD3 mouse monoclonal (Abnova #H00010714-M01, WB dilution 1:1000), anti-RFP rabbit polyclonal (GeneTex # GTX127897 WB dilution 1:5000), anti-BLM rabbit (kind gift from R. O'Sullivan lab) and anti-b-actin (Sigma #088M4804, WB dilution 1:10000) antibodies.

### Immunofluorescence and telomere fluorescence in situ hybridization

Cells were grown in 6-well plates on sterile glass coverslips. After treatments, cells were washed 2 times with ice-cold PBS 1X and incubated for 2 min with cytoskeletal extraction buffer (CSK:

10 mM PIPES, 100 mM NaCL, 300 mM sucrose, 2 mM $MgCl_2$, 0.25% triton, 1 mM DTT) on ice (for pRPA32 and RAD51 antibodies) and then fixed in 2% formaldehyde in cold PBS (PFA) for 15 min on ice, or ice-cold methanol/acetone (1/1) on ice for 10 min for PAR detection. After 3 washes in PBS 1X, cells were permeabilized for 10 min in PBS containing 0.5% triton and then blocked in blocking buffer (PBS supplemented with 1% BSA and 10% normal goat serum) for 1 h. Primary antibodies were diluted in blocking buffer and incubated overnight at 4 °C. Cells were washed 3 times in PBS containing 0.2% triton and incubated with secondary antibodies conjugated with Alexa Fluor 488, 594, 568, or 647 for 1 h at room temperature. Cells were washed 3 times in PBS containing 0.2% triton, rinsed in PBS, and water before being mounted on microscope slides in Diamond AntiFade containing Dapi (Invitrogen) at room temperature for at least overnight. For FISH staining, cells were dehydrated in successive baths of 70%, 90% and 100% ethanol after the water wash and coverslips were dried and mounted upside down on microscope slides containing 50 ml of hybridization buffer containing the telomere FISH PNA probe from PNA Bio (70% formamide, 10 mM Tris HCl pH 7.5, 1x Maleic Acid buffer, 1x MgCl2 buffer; TelC-Alexa488 1/200 (CCCTAAA)3). Slides were heated at 75 °C for 10 min and left to hybridize for 2 h at room temperature. Coverslips were returned to the 6-well plates to perform 2 15 min washes in hybridization buffer and 3 washes in PBS before mounting with Diamond AntiFade containing Dapi.

Antibodies used were anti-Poly(ADP-Ribose)10H mouse monoclonal (Enzo Life Science #ALX-802-220-R100, IF dilution 1:1000), anti-Poly/Mono-ADP Ribose (E6F6A) (Cell Signaling #83732, IF dilution 1:10000), anti-phospho RPA32 (S4/S8) rabbit polyclonal (Bethyl #A300-254A-T, IF dilution 1:500), anti-RAD51 rabbit monoclonal (Abcam #ab133534, IF dilution 1:1000).

### Proximity ligation assay PLA

DuoLink® In Situ Red Starter Kit Mouse/Rabbit (Sigma-Aldrich) was used to detect interacting proteins. Cells were fixed in 2% PFA for 10 min, and then were permeabilized for 10 min in PBS containing 0.5% triton and then were blocked in blocking buffer (PBS supplemented with 1% BSA and 10% normal goat serum) for 1 h. Samples were incubated at 4 °C overnight with specific primary antibodies to the proteins to be detected. Then slides were washed with 1X Wash Buffer A and

subsequently incubated with the two PLA probes diluted in antibody diluents for 1 h, then with the Ligation-Ligase solution for 30 min, and then with Amplification-Polymerase solution for 100 min in a pre-heated humidified chamber at 37 °C. Before imaging, slides were washed with 1X Wash Buffer B and mounted with Diamond AntiFade containing Dapi.

Inhibition of APE1 endonuclease in the PARP2/TRF2 PLA experiments was performed using the APE1 inhibitor III from Sigma (cat#262017). Cells were pre-treated with the APE1 inhibitor for one hour, before the MG2I dye addition.

Antibodies used were anti-TRF2 rabbit polyclonal (Novus Biologicals, NB110-57130, IF dilution 1:500), anti-PARP1 C-2-10 mouse monoclonal (Enzo Life Science #BML-SA249 BML-SA249, IF dilution 1:1000), anti-PARP-2 (4G8) mouse monoclonal (Enzo Life Science #ALX-804-639-L001 IF dilution 1:1000), anti-DYKDDDDK (FLAG) (FG4R) mouse monoclonal (Invitrogen #MA1-91878, IF dilution 1:1000), anti-POLD3 mouse monoclonal (Abnova #H00010714-M01, IF dilution 10 mg/mL), anti-RFP rabbit polyclonal (GeneTex # GTX127897 IF dilution 1:1000), anti-Poly/Mono-ADP Ribose (E6F6A) (Cell Signaling #83732, IF dilution 1:10000).

### Chromosome metaphase spreads and fluorescence in situ hybridization (FISH)

Chromosome metaphase spreads were prepared as previously described in ref. 42. Briefly, cells were harvested in conditioning media after treatment with 0.05 mg/ml of colcemid for 2 h. Cell pellets were resuspended, incubated at 37 °C for 8 min in 75 mM KCl, and fixed in a fixative solution (methanol and glacial acetic acid 3:1 ratio). Fixed cells were dropped on microscope slides and left to dry for at least 24 h in the dark. Slides were then incubated in 2% formaldehyde, and cells were treated with RNaseA and Pepsin at 37 °C and dehydrated in successive baths of 70%, 90%, and 100% ethanol. A hybridization mixture containing TelC-Alexa488 and Cy5-Pan-centromere probes is then added, and the slides heat up at 75 °C for 10 min followed by incubation at room temp in a hybridization chamber for 2 h or at 4 °C overnight. Slides are washed with hybridization buffers A (70% (v/v) formamide deionized,10 mM Tris-HCl, pH 7.5) and B (50 mM Tris-HCl, pH 7.5, 50 mM NaCl 0.8% (v/v) Tween 20) and mounted in Diamond AntiFade containing Dapi (Invitrogen) at room temperature for at least overnight.

### Mitotic DNA synthesis assay and EdU labelling

After 18 dye and light treatments, cells were allowed to recover for 8 h prior to an 8 h incubation with 7 mM Cdk1 inhibitor RO3306 (Millipore). Cells were washed twice with PBS and incubated in fresh media containing 1X EdU and 50 ng/ml concentrations colcemid for 1 h prior to harvest. Metaphase spreads were prepared as described above and EdU staining was performed using Click-iT™ EdU Alexa Fluor™ 594 imaging kit (ThermoFisher) prior to telomere FISH staining.

### Chromosome orientation FISH (CO-FISH)

The procedure for telomere CO-FISH was performed following the protocol from[47], with minor modifications. Briefly, after the last treatment of the chronic (N18), cells were left to recover for 8 h, and then they were incubated with 7.5 µM BrdU and 2.5 µM BrdC for 16 h before harvesting. Metaphase spreads were prepared as described above. After rehydration with 1x PBS, the newly replicated strands were degraded using treatment with 0.5 mg/mL RNase A for 15 min at 37 °C, stained with 0.5 µg/mL Hoechst 33258 in 2X SSC for 15 min at room temperature, and exposed to 365 nm UV light for $5.4 \times 10^3$ J/m², followed by treatment with 100 µL of 10 U/µL exonuclease III (NEB) for 1 h at 37 °C. Slides were then washed with PBS and serially dehydrated as above. Cells were hybridized with TelC-Alexa488 (PNA Bio) probe for 2 h at room temperature, rinsed twice with wash buffer A (70% (v/v)

formamide deionized,10 mM Tris-HCl, pH 7.5), and three times with wash buffer B (50 mM Tris-HCl, pH 7.5, 50 mM NaCl 0.8% (v/v) Tween 20) and hybridized with TelG-AlexaCy3 (PNA Bio) probe for 2 h at room temperature. Cells were washed again as described above and mounted in Diamond AntiFade containing Dapi (Invitrogen) at room temperature for at least overnight.

### Protein purification
Human DNA polymerase δ was expressed in *E. coli* (BL21-CodonPlus (DE3)-RP) and purified by conventional column chromatography, as previously described[75]. Briefly, bacteria cells were grown in Terrific broth supplemented with ampicillin, streptomycin and chloramphenicol at 15 °C until culture reached an A600 value of 0.6. IPTG was then added, and incubation continued for 15 h. The cells lysed by addition of 333 μl buffer containing 500 mM NaCl, 100 mM spermidine, 4 mg/ml lysozyme and 1 mM phenylmethylsulfonyl fluoride, per 1 g cells. After adding imidazole to 5 mM, lysate (up to 40 ml) was applied at 1 ml/min to a 5-ml HiTrap chelating HP column which had been treated with 0.1 M NiSO$_4$ and then equilibrated with buffer A containing 500 mM NaCl and 5 mM imidazole. The column was washed with 50 ml of equilibration buffer and eluted with 50 ml of a linear gradient of 5-100 mM imidazole in buffer A containing 500 mM NaCl. Binding is due to an affinity of the p66 subunit for the resin, even in the absence of any artificial tags. Fractions containing Polδ were pooled, diluted with buffer A to 400 mM of NaCl, and applied at 0.5 ml/min to a 5-ml HiTrap heparin HP column equilibrated with buffer A containing 400 mM NaCl. The column was washed with 50 ml of equilibration buffer, and eluted with 50 ml of a linear gradient of 400–800 mM NaCl in buffer A. Fractions containing all of the subunits of Polδ eluting at about 570 mM NaCl were pooled and concentrated using an centrifugal filter device, Amicon Ultra-4 100,000 MWCO (Millipore), and applied at 0.1 ml/min onto a Superose 6 HR 10/30 column equilibrated with buffer A containing 500 mM NaCl. All subunits of Polδ co-eluted and the peak fraction was frozen in liquid nitrogen and stored at −80 °C.

### POLD3 heteromodification assay
125 nM of purified Polδ was incubated with or without 10 ng of hPARP2 in 1X activity buffer (10X buffer: 500 mM Tris–HCl pH 8.0, 40 mM MgCl2, 2 mM DTT, 500 μg/ml BSA) for 5 min, at RT. The reaction was started by adding 125 nM of activated DNA (Enzo) and 2.5 μM of NAD$^+$ (Enzo) and carried out for 30 min at 25 °C. The reaction was stopped by adding an equal volume of 2X Laemmli sample buffer (BioRad) with 0.1 M EDTA. Samples were boiled for 5 min at 100 °C and run on a 10% polyacrylamide SDS-PAGE gel at 80–120 V for ~1.5 h. The gel was transferred onto a nitrocellulose membrane at 25 V for 1.5 h at RT. The membrane was blocked in 5% BSA in TBST and incubated with the rabbit anti-Poly/Mono-ADP Ribose (E6F6A) (Cell Signaling #83732) at 1:1000 concentration overnight at 4 °C. The next day, the membrane was incubated with a goat anti-rabbit secondary antibody at 1:5000 concentration for 1 h at RT and developed with ECL detection reagents (Amersham/Cytiva).

### RFP immunoprecipitation
PARP1/2KO cells were seeded in normal growth media, so they were 70–80% confluent at the time of transfection. Cells were transfected with POLD3-RFP, using Lipofectamine 3000 as described above. Cells were harvested and the pellet was resuspended in 200 μL of ice-cold RIPA buffer supplemented with DNaseI 75 Kunitz U/mL, MgCl2 2.5 mM, PMSF 1 nM, and with Roche protease inhibitor cocktail tablets 1X for 30 min on ice and then centrifuged at maximum speed in a microfuge for 15 min at 4 °C. The cleared lysate (supernatant) was transferred to a precooled tube and 300 μL of Dilution buffer (10 mM TrisHCl pH 7.5, 150 mM NaCl, 0.5 mM EDTA) supplemented with 1 mM PMSF and protease inhibitor cocktail was added. The extracts (for a total of 500 μL) were incubated with 25 μl magnetic RFP binding protein beads

(RFP-Trap-M, Chromotek) at 4 °C for 1 h. After incubation, the beads were washed three times with the wash buffer (10 mM Tris/Cl pH 7.5, 150 mM NaCl, 0.05 % Nonidet™ P40 Substitute, 0.5 mM EDTA), resuspended in 12 μl of Laemmli buffer, and heated for 3 min at 100 °C and run on a 10% polyacrylamide SDS-PAGE gel at 80–120 V for ~1.5 h. The gel was transferred onto a nitrocellulose membrane at 25 V for 1.5 h at RT. The membrane was blocked in 5% milk in TBST and incubated with the mouse anti-PARP2 antibody (Enzo) at 1:1000 concentration overnight at 4 °C. The next day, the membrane was incubated with a goat anti-mouse secondary antibody at 1:5000 concentration for 1 h at RT and developed with ECL detection reagents (Amersham/Cytiva). Anti-POLD3 (Abnova) antibody has been used to confirm POLD3 immunoprecipitation efficiency.

### Microscopy quantification
The number of PAR, pRPA32, and RAD51 foci colocalizing with telomeres, and PLA foci were measured using NIS Element Advanced Research software (Nikon) after deconvolution. The process included the isolation of individual nuclei as regions of interest (ROI) using DAPI channels. The intensity tool was used to select the foci in the appropriate channel. The threshold was maintained for all images within the same replicate experiments. The foci were then qualified as "objects" and automatically quantified by the software for each ROI selected. The number of PAR, pRPA32, RAD51, and PLA foci and intersections per ROI were exported to Excel for data batch analysis using RStudio (open source) and then imported into GraphPad Prism 9 for graphing and statistical analyses.

### Reporting summary
Further information on research design is available in the Nature Portfolio Reporting Summary linked to this article.

## Data availability
All data generated or analyzed during this study are included in this published article and its supplementary information files. Source data are provided with this paper.

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

## Acknowledgements

E.F. laboratory is supported by an NIH MIRA R35 award (R35GM142982) and Start-up fundings from UPMC Hillman Cancer Center. We thank Drs. Ryan Barnes and Roderick O'Sullivan for critical reading of the manuscript and helpful discussions.

## Author contributions

D.M. performed all the experiments. N.L. established the PARP2KO cell line. R.L.D and M.H. purified the polymerase delta complex. S.D.L and C.C. provided technical support. S.U. conducted statistical analyses and provided codes for microscope imaging analyses. E.F. conceived the study. E.F. and D.M. wrote the manuscript.

## Competing interests

The authors declare no competing interests.
