## [Peer Review File · Nature Communications]

PARP2 promotes Break Induced Replication-mediated telomere fragility in response to replication stressREVIEWER COMMENTS

Reviewer #1 (Remarks to the Author):

Summary of paper:

In this manuscript, the authors aim to understand the role of PARP2 at telomeres following chronic induction of telomeric oxidative stress. The authors use a well-established chemoptogenetic system, known as FAP. The FAP peptide complexes with the MG2i photosensitizer dye, which when exposed to 660nm light, generates a single oxygen radical, triggering the formation of 8-oxo-G lesions. When FAP is fused to TRF1, it localises the oxidative damage to the telomeres. The authors show that depletion of PARP1 in the context of oxidative damage induction causes an increase in fragile telomeres, and that PARP2 is recruited to telomeres after the induction of oxidative damage. While PARP1 protects cells from fragile telomeres and telomere loss, PARP2 promotes telomere fragility and loss. The major observation in this study is that loss of PARP2 in PARP1 depleted cells rescues the telomere fragility phenotype. The authors claim that PARP2 contributes to BIR and MiDAS in PARP1 KO cells, and further show that the ADP-ribosyl-transferase catalytic activity of PARP2 is important in the induction of telomere fragility and telomere BIR, but this later appears to be contradicted when the catalytic mutant of PARP2 can still recruit and retain POLD3 at the telomeres.

Specific questions and comments:

1. Fig 1b - there is a white line through the representative image panel.
2. Lines 142-144: 'PARP1 has a broader substrate specificity and can recognise other types of oxidative lesions...' – this claim is too much of a leap based on the very small difference in the mean TRF2/PARP1 and TRF2/PARP2 PLA foci of H₂O₂ treated vs dye and light treated (telomere specific damage). PARP1 and PARP2 ChIP should be performed to assess telomere vs genomic recruitment. e.g. with telomere and Alu-probes.
3. Minor general comment: many of the experiments are performed in HeLaFAP. Are the findings of PARP2 applicable to other forms of more naturally derived oxidative stress? Would like to see more of the experiments repeated with H₂O₂.
4. Lines 173-174 referring to Fig 1G: would like to see the actual growth curves. How does the growth slope compare between dye + light treated HeLaFAP to PARP1 KO, PARP2 KO and PARP1/2 KO? Is there synergy when both PARP proteins are knockout? Is there an explanation for such synergy?
5. Fig 1L: No representative microscopy images to support the quants.
6. Is there a correlation between telomere length and sensitivity to PARP2 loss?
7. Lines 197-201: Should be Fig 1K and 1L, not Fig 3A.
8. Line 203: Typo in depletion – should be depletion. Figure 2 title has this typo too.
9. Lines 209-212 referring to Fig 2C-F: would benefit from the inclusion of representative images to show changes in the amount of EdU incorporation at single and both chromatids in different treatments (PARP1 KO, HeLaFAP, PARP2 KO, etc).
10. Lines 221-237 referring to Fig 2G-I: Is there another way to support the role of BIR besides using CO-FISH, such as staining for POLD3 (or other BIR factors) foci at the telomeres? The strength of this conclusion warrants further demonstration.
11. Figure 2I: What happens to BIR if PARP2 was knocked out together with PARP1? The legend of Fig 2I says HeLaFAP, PARP1/2KO and PARP2 KO were all quantified but the graph only shows PARP1 KO. Given the purpose of this figure is to show how PARP2 KO prevents BIR in PARP1 KO, it is critical to show the PARP1/2 KO data.
12. Lines 287-292 referring to Fig 4A-B: The representative images are too small and too poor resolution to be supportive of the quants. The quants themselves are very low and the changes small. I do not find this figure very informative/convincing. Would also benefit from EdU-RPA-telomere triple colocalizations.
13. Lines 310-311 referring to figure 5A: How does the POLD3/TRF2 PLA of PARP1/2 KO compare to PARP2 KO and HeLaFAP? The authors make the point that PARP1/2 KO has fewer PLA and hence recruitment of POLD3 to telomeres, but is this less than the WT HeLaFAP cells and do they see the same level of PLA with the PARP2 KO as with the double knockout?

14. Lines 312-323 referring to Figure 5A-E: This is in regard to the increase in POLD3/TRF2 PLA observed with the PARP2 mutant. If PARP2 recruits POLD3 to the telomeres, you would expect PARP2 catalytic mutant to be unable to recruit POLD3. So how does the mutant retain POLD3 at the telomeres if it cannot be recruited in the first place? Alternatively, if the PARP2 catalytic domain is not required for POLD3 recruitment, then what domain/activity of PARP2 is recruiting POLD3 to the telomeres? This warrants further discussion/explanation.
15. PARP2 KO does not impair telomere MiDAS under conditions of lesion induction. How is this reconciled with induction of telomere MiDAS upon PARP2 expression and decrease in fragile telomeres after CO-FISH staining, which the authors say is indicative of conservative synthesis?
16. Impaired telomere MIDAS after expression of the PARP2 E558A compared to WT (Fig. 3G-J) indicates an impaired function of POLD3 at telomeres, but this needs to be addressed. Why does MIDAS not increase with increased POLD3 at telomeres after expression of the E558A PARP2 mutant?
17. The TRF of PARP1 and/or PARP2 KO over long-term passage should be moved to the main Figure.
18. Lines 312-323 referring to Figure 5A-E: The images, especially in 5D, show PLA staining well outside of the nucleus. Are these also POLD3/PARP2 PLA? What is the explanation for this?
19. Line 337: the text does not explain why BLM depletion would increase POLD recruitment.

Reviewer #2 (Remarks to the Author):

In this report, the authors suggest that PARP2 plays a unique role during break-induced replication at telomeres that is dependent on the presence and activity of PARP2 by targeting PolD3. However, the study ends with analysis of DNA synthesis during BIR but is not designed to be telomere specific, whereas the first 4 figures are specific to telomeric DNA damage.

Other concerns include:

Page 5, Results section: Authors have not documented the APE1 inhibitor used. The explanation for the APE1i effect is stated as "This effect can be attributed to the ability of PARP1 to compete with APE1 for the binding of AP site" – This statement should be clarified.

Page 5, Results section: Not clear on the statement "Single-cell clone populations ...". Did the authors select single-cell clones or populations?

Page 6: Fig 3A does not show the BLM siRNA immunoblot results as indicated. Should be Fig 1K.

Page 8: What is meant by the phrase "the PARP2 catalytic mutant in fusion with a flag tag"?

Throughout: "Fig.s" should be changed to "Figs."

Fig 3A: Authors should also include a direct comparison of the WT cell line to ensure that the level of expression of the transgene is similar to the endogenous level of expression for PARP2.

Page 9: The phrase "... MiDAS arose dependently of PARP2 activity (Fig. 3)" should be corrected to read as "... MiDAS arose dependent on PARP2 activity (Fig. 3)".

Fig 5F: This data shows that PARP2 can modify PolD3 when isolated. Authors should also show that PolD3 is adp-ribosylated when cell are treated with ROS or the FAP.

Top of page 5: The authors indicate "... antibody revealed that both PARP1 and PARP2 are efficiently recruited to telomeres immediately after acute dye and light exposure (Fig.s 1B and 1C)." but also should lest Fug 1D.

Fig 1E and throughout: It is expected that at least two KO clones are to be evaluated.

Bottom of page 5: The authors indicate "...fragility compare to other cell lines (Fig. S1F),..." but this should be corrected to read "... fragility compared to other cell lines (Fig. S1F), ...".

Fig 1G: Is the decreased growth in the PARP1/PARP2-dKO additive as compared to the PARP1-KO and PARP2-KO or synergistic?

Fig S1I: The table should be labeled.

Fig S1J: Why were the individual KOs not evaluated. This should be included.

Fig 3 (bottom of page 6): I note that this figure is mentioned after Figure 1 is presented but before Figure 2 is presented in the Results section. However, it appears the authors meant to refer to Fig 1L (not Fig 3) and it is also noted that Fig 1K is not mentioned in the text.

Fig 2I: Why were the other two cell lines (PARP2-KO and PARP1/2-KO) not evaluated for fragile telomere per metaphase? This should be included.

The authors should also evaluate PARP1/2 inhibitors vs PARP1 selective inhibitors.

Figs 3B-J: Does re-expression of PARP1 give the same effect as re-expression of PARP2?

Fig 4: The cells labeled as "FLAG PARP2" and E558A", are these the PARP1/2-KO complemented cells? If so this should be clarified.

Bottom of page 9: The statement "... demonstrated that telomere fragility and MiDAS arose dependently of PARP2 activity..." should be corrected to read "... demonstrated that telomere fragility and MiDAS is dependent on PARP2 activity ...".

Fig 5: Authors should show that PolD3 is PARylated in cells under the conditions used herein for replication stress.

Page 11: What is meant by the phrase "requires a direct interaction with PolD3 and its PARylation"?

The authors use the designation 'ko' for knock-out but the term normally used is 'KO'. Also, this should be defined.

Throughout: Some English language editing is suggested.

Reviewer #3 (Remarks to the Author):

This manuscript elucidates the role of PARP2 in fostering telomere fragility and averting telomere loss during replication stress. It proposes that PARP2 regulates telomere fragility through two stages of the break-induced replication (BIR) pathway: DNA end resection and DNA synthesis. The findings indicate that while the DNA end resection phase of BIR is not contingent on PARP2 catalytic activity, PARP2 aids BIR-dependent mitotic DNA synthesis through the PARylation of the Pol delta subunit POLD3. The paper contends that PARP2 is central to managing replication stress at telomeres and that it can induce telomere fragility through the BIR pathway. This study is compelling given its exploration of further roles of PARP2 in response to replication stress. However, the representation of dysfunctional phenotypes indicates that PARP2's role in mediating replication stress might be negligible, lessening the impact of the study. Additionally, the evidence for PARP2's role in prompting telomere fragility

through PARylation-independent and -dependent steps of the BIR pathway lacks potency.

Major Points:

1. Figures 1B through 1D imply greater efficiency in the recruitment of PARP1 to telomeres in comparison to PARP2 during replication stress. Furthermore, chronic exposure treatment decelerates the expansion of PARP1-KO cells, while the proliferation rate of PARP2-KO cells remains steady (Fig. 1G). These findings strengthen the premise that PARP1 is the primary factor in managing replication stress, a theory reinforced by substantial previous studies. Conversely, PARP2 might have a secondary role in replication stress regulation.
2. Lines 160-161 in the manuscript conclude that findings suggest a role of PARP2-dependent PARylation upon exposure to a dye and light. However, no PAR signal is detected at telomeres post the treatment in PARP1-KO cells expressing PARP2, suggesting that PARP2 doesn't contribute to the major PARylation events during replication stress. Hence, the presented conclusions lack support from the results. Moreover, the authors propose that PARP2 advances BIR-dependent mitotic DNA synthesis through PARylation of the Pol delta subunit POLD3, suggesting an expected PARylation signal in PARP1-KO cells expressing PARP2 under replication stress. This inference contradicts the results demonstrated in Figures S1D and S1E.
3. The authors illustrate how PARP2 governs telomere fragility using PARP1/2-KO cells complemented with either wild or mutant type PARP2. However, the PARP2-WT or Mutant proteins in these cells are overexpressed, as evinced by Figure S3A. A similar magnitude of PARylation level in the re-expressed PARP2-WT-PARP1/2-KO cells as that in the PARP1-KO cells expressing endogenous PARP2 is presumable. However, the PARylation level in the re-expressed PARP2-WT-PARP1/2-KO cells is excessively high, even comparable to parental HeLa cells. Hence, this indicates an overexpression of PARP2 in the re-expressed PARP2-WT-PARP1/2-KO cells. Since most known functions of PARP2 duplicate those of PARP1, the overexpressed PARP2 could potentially imitate the functions of PARP1 rather than performing as an authentic PARP2 entity. Hence, the evidence with FLAG-PARP2 complemented in PARP1/2KO cells appears less persuasive. Before further consideration, the authors should select cell clones exhibiting comparable endogenous PARP2 expression and reiterate all the experiments.
4. The Western blot in Figure 5F fails to convincingly signify POLD3 as a likely PARP2 substrate. For enhanced evaluation, the Western blots should include molecular weight markers. Moreover, the observed PAR signal might stem from PARylated-PARP2 rather than PARylated-POLD3. The authors should conduct mass spectrometry to verify if POLD3 is indeed a potential PARP2 substrate. Furthermore, an investigation of whether POLD3 is a substrate of PARP1 rather than PARP2 should be undertaken.

Minor Points:

1. To indicate their precise position, the Western blots should incorporate dashed lines at the positions of molecular weight markers.
2. Proper presentation of the references cited in lines 129 and 148-149 is necessary.
3. The image quality of Figure 1A, especially the text, needs improvement.
4. The study should provide p-values for Figures S3C and S3D.
5. The text contains numerous mislabeling, for instance, there is a reference to "Fig. 7" in line 379. However, "Fig. 7" doesn't exist in the manuscript.

REVIEWER COMMENTS

Reviewer #1 (Remarks to the Author):

Summary of paper:

In this manuscript, the authors aim to understand the role of PARP2 at telomeres following chronic induction of telomeric oxidative stress. The authors use a well-established chemoptogenetic system, known as FAP. The FAP peptide complexes with the MG2i photosensitizer dye, which when exposed to 660nm light, generates a single oxygen radical, triggering the formation of 8-oxo-G lesions. When FAP is fused to TRF1, it localises the oxidative damage to the telomeres. The authors show that depletion of PARP1 in the context of oxidative damage induction causes an increase in fragile telomeres, and that PARP2 is recruited to telomeres after the induction of oxidative damage. While PARP1 protects cells from fragile telomeres and telomere loss, PARP2 promotes telomere fragility and loss. The major observation in this study is that loss of PARP2 in PARP1 depleted cells rescues the telomere fragility phenotype. The authors claim that PARP2 contributes to BIR and MiDAS in PARP1 KO cells, and further show that the ADP-ribosyl-transferase catalytic activity of PARP2 is important in the induction of telomere fragility and telomere BIR, but this later appears to be contradicted when the catalytic mutant of PARP2 can still recruit and retain POLD3 at the telomeres.

Specific questions and comments:

1. Fig 1b - there is a white line through the representative image panel.

We apologize for what appears to be an issue with the conversion of the ai file into a reduced size PDF. This line does not appear on the high-quality image anymore.

2. Lines 142-144: 'PARP1 has a broader substrate specificity and can recognise other types of oxidative lesions...' – this claim is too much of a leap based on the very small difference in the mean TRF2/PARP1 and TRF2/PARP2 PLA foci of H₂O₂ treated vs dye and light treated (telomere specific damage). PARP1 and PARP2 ChIP should be performed to assess telomere vs genomic recruitment. e.g. with telomere and Alu-probes.

The goal of this initial experiment is to assess whether both PARP1 and PARP2 are first-responders to oxidative lesion induction at telomeres. Our data indicate that both PARP enzymes are efficiently recruited to the telomeres right after dye and light treatment indicating that both are involved in the first response of the cells to 8oxoG induction.

The use of H₂O₂ treatment was to provide a positive control to our experimental design rather than comparing the efficiency of PARP1 and PARP2 recruitment to lesions induced by H₂O₂ to lesions induced by the FAP tool. H₂O₂ is a broad DNA damaging agent that induces DNA base oxidation, including but not restricted to 8oxoG, as well as DNA breaks in the whole genome including, but not restricted to, telomeres. In past studies, H₂O₂ has been shown to efficiently recruit both PARP1 and PARP2 to DNA damage sites (Ame et al., 1999; Hanzlikova et al., 2017). Accordingly, we observed formation of PLA foci between TRF2 and PARP1 or PARP2 antibodies, regardless of the treatment used.

The past literature has also reported the large difference between PARP1-specific and PARP2-specific DNA substrates. While PARP1 can recognize DNA breaks harboring any ends as well as non-B DNA structures (Laspata et al., 2023), PARP2 has the highest affinity for breaks harboring phosphorylated ends such as the ones generated by the bi-functional glycosylase OGG1 when acting 8oxoG lesions (Svilar et al., 2010). Because H₂O₂ does not trigger 8oxoG only and is not targeting telomeres exclusively, our data showing that PARP2 forms less PLA foci when cells are treated with H₂O₂ than when they are subjected to the 8oxoG-specific FAP tool, illustrates these

past observations. Conversely, because PARP1 has a broad substrate affinity, it is expected that H₂O₂ will trigger 8oxoG, other oxidative lesions and direct breaks at telomeres that will all trigger PARP1 recruitment which is illustrated by our data.

Our conclusion that “**PARP1 has a broader substrate specificity and can recognize other types of oxidative lesions**” is therefore based on previous work and our data only confirm these observations. To clarify these points, we have rephrased the paragraph.

Finally, while it would be interesting to perform ChIP assays as suggested by the reviewer, none of the PARP2 antibodies commercially available are suitable for such assay.

3. Minor general comment: many of the experiments are performed in HeLaFAP. Are the findings of PARP2 applicable to other forms of more naturally derived oxidative stress? Would like to see more of the experiments repeated with H₂O₂.

In this study, our goal is to study the role of PARP2 in the response to replication stress. We have chosen to trigger replication stress using 2 different methods: through oxidative DNA lesion using the FAP tool and by knocking down the BLM helicase to increase the level of G4s.

We have previously demonstrated that chronic induction of 8oxoG lesions triggers replication stress at telomeres (Fouquerel et al., 2019). Because replication stress at telomeres leads to telomere fragility and loss that are easily visualized on metaphase spreads, telomeres offer a convenient way to measure replication stress.

Our FAP tool generates singlet oxygen which is a type of ROS naturally found in cells and induces 8oxoG lesions which are the most common oxidative DNA lesions. The FAP tool provides us with a way to deliver ROS and induce 8oxoG locally. It is likely that other forms of oxidative stress can similarly trigger replication stress at telomeres. However, H₂O₂, for instance, induces a variety of DNA lesions beyond 8-oxo-G and impacts the whole genome as much as other macromolecules in the cells, leading to a broad variety of molecular responses. These were the reasons why in 2016 we set out to develop the FAP tool which allowed us to attribute the overall genomic instability to the sole telomeric 8-oxoG lesions (Fouquerel et al., 2019). Using H₂O₂ to repeat our experiments would therefore not be relevant to understand the impact of oxidative DNA-damage-dependent replication stress at telomeres.

4. Lines 173-174 referring to Fig 1G: would like to see the actual growth curves.

We have chosen to display graphs representing the slopes of the growth curves obtained for each replicate to better illustrate the reproducibility of our data. The curves themselves are presented in supplemental figure 2, panel a.

How does the growth slope compare between dye + light treated HeLaFAP to PARP1 KO, PARP2 KO and PARP1/2 KO? Is there synergy when both PARP proteins are knockout? Is there an explanation for such synergy?

We have extended our statistical analysis of our data (now Figure 2b). As previously noted, PARP1 depletion impacts cell growth of treated cells but not PARP2 depletion. However, depletion of both enzymes has a significantly greater impact on cell growth than upon their individual depletion. We explain this synergy by a role of both enzymes in promoting the response to replication stress via distinct pathways which our subsequent data illustrate. We have added these observations in the main text.

5. Fig 1L: No representative microscopy images to support the quants.

We apologize for omitting to display representative images. We have added them in the new supplementary figure 2i.

6. Is there a correlation between telomere length and sensitivity to PARP2 loss?

We thank the reviewer for this very interesting question. To address this, we have selected 1 more clone per cell line with a different mean telomere length than the clone initially used (see new data displayed in Supplementary figure 2d-f). We have exposed these additional clones to chronic dye and light treatments as described in the new Figure 2a and repeated the following experiments:

- Metaphase spread followed by FISH to quantify the number of fragile telomeres and signal-free ends (new Supplementary figure 2g and 2h)
- Metaphase spreads after EdU pulse treatment to quantify BIR-dependent conservative DNA synthesis or MiDAS (New Supplementary figure 4a-c).
- Metaphase spreads followed by CO-FISH staining to determine whether BIR-dependent DNA synthesis is responsible for the fragile phenotype (New Supplementary figure 5e-g).
- PLA for POLD3:TRF2 (New Supplementary figure 7b)
- Recruitment of Rad51 at telomeres following chronic oxidative stress (new Supplementary figure 6e).

In summary, our data indicate that regardless of the size of telomeres, PARP2 loss in PARP1ko cells prevents telomere fragility and PARP2 complementation in the PARP1/PARP2ko cells restores this phenotype. Similarly, the variation in the size of the bulk telomeres did not impact our conclusion that PARP2 drives BIR.

We invite the reviewer to refer to the new manuscript for more details about the new data collected.

7. Lines 197-201: Should be Fig 1K and 1L, not Fig 3A.

corrected

8. Line 203: Typo in delpletion – should be depletion. Figure 2 title has this typo too.

corrected

9. Lines 209-212 referring to Fig 2C-F: would benefit from the inclusion of representative images to show changes in the amount of EdU incorporation at single and both chromatids in different treatments (PARP1 KO, HeLaFAP, PARP2 KO, etc).

This is a good suggestion and we have added representative images to illustrate the Edu staining on multiple chromosomes within the same metaphase spreads (see new Figure 4c).

10. Lines 221-237 referring to Fig 2G-I: Is there another way to support the role of BIR besides using CO-FISH, such as staining for POLD3 (or other BIR factors) foci at the telomeres? The strength of this conclusion warrants further demonstration.

In the previous Figure 2, we quantified two features of BIR related to the DNA synthesis step of this pathway: CO-FISH but also MiDAS. Both assays support the role of BIR.

In the previous Figures 4 and 5, we have followed the recruitment of the BIR factors RPA, Rad51, and PolD3 and concluded that end resection is independent of PARP2 activity while the DNA synthesis steps rely on ADP-ribosylation. Following this comment and comment #12 below, we have removed the data presented previously in Figure 4 and have re-organized our data as follows:

Our new Figure 5 now displays our CO-FISH data along with data showing the recruitment of RPA and RAD51 upon both BLM depletion and chronic dye and light treatment. Then our new Figure 6 shows recruitment of POLD3 to telomeres in the same treatment conditions. Our new data strengthen our conclusion of a role for BIR in the occurrence of fragile telomeres dependently of PARP2. We invite the reviewer to refer to the newly edited article for more details.

11. Figure 2I: What happens to BIR if PARP2 was knocked out together with PARP1? The legend of Fig 2I says HeLaFAP, PARP1/2KO and PARP2 KO were all quantified but the

graph only shows PARP1 KO. Given the purpose of this figure is to show how PARP2 KO prevents BIR in PARP1 KO, it is critical to show the PARP1/2 KO data.

In this figure, we demonstrate the occurrence of BIR events using 2 techniques: MiDAS and CO-FISH.

We have done these experiments using all our cell lines and while we have presented the MiDAS data for the 4 cell lines in the main figure (now displayed in new Figure 4) we have only presented CO-FISH data (now displayed in Figure 5) for the PARP1ko cell line in the main figure and added the results obtained with the other cell lines in the Supplemental Figure. We apologize for omitting to correct the legend accordingly after a change made prior to submission. We have now edited it. Because HeLaFAP, PARP2KO, and PARP1/2KO cells do not trigger fragile telomeres, CO-FISH would not display any change with the treatment. Therefore, we decided to keep these data in the new Supplemental Figure 5 to keep our most important ones highlighted.

12. Lines 287-292 referring to Fig 4A-B: The representative images are too small and too poor resolution to be supportive of the quants. The quants themselves are very low and the changes small. I do not find this figure very informative/convincing.

We apologize for our small images as we were attempting to illustrate our observations with our 6 cell lines using 2 experimental conditions and testing 3 BIR-related steps. Our work following the divers' comments (see our response to comment #10 above) from the reviewers pushed us to make the decision to remove these data, performed upon acute exposure, from our article. They are now replaced by data showing recruitment of RPA and Rad51 at telomeres upon chronic oxidative damage induction as well as upon BLM depletion (see new Figure 5). Images for the graphs related to Rad51 are displayed with enlargements in the new Supplementary Figure 6.

Would also benefit from EdU-RPA-telomere triple colocalizations.

BIR pathway is a DNA repair pathway during which RPA coating of ssDNA is one of the first steps. DNA synthesis is a downstream step that occurs after RPA has been replaced by Rad51. The timing of these events would not allow for a triple colocalization between EdU-RPA-telomeres.

13. Lines 310-311 referring to figure 5A: How does the POLD3/TRF2 PLA of PARP1/2 KO compare to PARP2 KO and HeLaFAP? The authors make the point that PARP1/2 KO has fewer PLA and hence recruitment of POLD3 to telomeres, but is this less than the WT HeLaFAP cells and do they see the same level of PLA with the PARP2 KO as with the double knockout?

We have now dedicated an entire figure to address this comment (Figure 6). We have re-done the PLA experiments in all our cell lines after chronic oxidative stress induction (Panels a and b) and BLM siRNA treatment (Panels c and d). Our data show that replication stress induction triggers a large increase of POLD3/TRF2 PLA foci in PARP1KO cells and in PARP1/2KO cells complemented with PARP2-FLAG and the E558A mutant. Only a small number of PLA foci were observed in HeLaFAP and PARP2KO cells but not in PARP1/2KO cells. However, these small increases were not observed upon BLM depletion. Our data suggest that POLD3 presence at telomeres requires PARP2 but not its activity.

14. Lines 312-323 referring to Figure 5A-E: This is in regard to the increase in POLD3/TRF2 PLA observed with the PARP2 mutant. If PARP2 recruits POLD3 to the telomeres, you would expect PARP2 catalytic mutant to be unable to recruit POLD3.

So how does the mutant retain POLD3 at the telomeres if it cannot be recruited in the first place? Alternatively, if the PARP2 catalytic domain is not required for POLD3 recruitment, then what domain/activity of PARP2 is recruiting POLD3 to the telomeres? This warrants further discussion/explanation.

We thank the reviewer for this comment that we are addressing with comment #16 as they are related to each other:

16. Impaired telomere MIDAS after expression of the PARP2 E558A compared to WT (Fig. 3G-J) indicates an impaired function of POLD3 at telomeres, but this needs to be addressed. Why does MIDAS not increase with increased POLD3 at telomeres after expression of the E558A PARP2 mutant?

We agree with the reviewer that this data is very unexpected, and it also surprised us. Based on previous work, we discussed our hypothesis in the Discussion section. In sum, in recent study from the Ying Liu lab (Wu et al., 2023, Nat Comm) demonstrated that both translesion polymerase Pol ζ and polymerase Pol δ are required to promote MiDAS and that they work together sequentially. Pol ζ seems to act upstream Pol δ , itself recruited thanks to an interaction of the REV1 interacting region (RIR) of POLD3 with REV1 and via a polymerase switch mechanism that remains unclear. Interestingly, POLD3 RIR comprises 2 residues that were found to be PARylated (Larsen et al., 2018) and Pol ζ and Pol δ share POLD3 as an accessory subunit. One hypothesis that we can propose is that ADP-ribosylation of RIR allows for the disruption of POLD3's interaction with PARP2 and its subsequent recruitment by REV1 which would enable the polymerase switch for the completion of DNA synthesis. Of course, this remains to be demonstrated and we are planning to focus our research effort on deciphering this mysterious polymerase switch that orchestrates the MiDAS mechanism.

15. PARP2 KO does not impair telomere MiDAS under conditions of lesion induction. How is this reconciled with induction of telomere MiDAS upon PARP2 expression and decrease in fragile telomeres after CO-FISH staining, which the authors say is indicative of conservative synthesis?

Telomere MiDAS do not occur in PARP2ko cells because PARP1 is still expressed in these cells. PARP1 is the main first responder in the repair of oxidative lesions and its presence is sufficient to repair 8oxoG and prevent the replication stress that would otherwise require BIR and MiDAS to occur. Moreover, as discussed in our article, work from the de Lange lab has also demonstrated that PARP1-dependent alt-EJ competes with BIR at telomeres under replication stress (Yang et al., 2020) which is in agreement with our data.

The re-expression of PARP2 (PARP2-FLAG) was performed in cells initially depleted in both PARP1 and PARP2. Re-expression of PARP2 was performed in these cells to verify the hypothesis that PARP2 mediates the response to replication stress that occurs in PARP1ko cells and orchestrates the occurrence of the fragile phenotype. In line with this, we observed a restoration of telomere fragility (New Figures 3e and 3g) and an increase of MiDAS events (Figure 4) that are indicative of BIR. Finally, a decrease in the number of fragile telomeres in a CO-FISH experiment is indicative of BIR-dependent DNA synthesis, further demonstrating the role of PARP2 in driving BIR (Figure 5).

To clarify these points, we have reworked the results section pertaining to these data and added elements in the discussion section as well as on our working model (please see newly added text and new figures 3,4,5 and 8).

16. Impaired telomere MIDAS after expression of the PARP2 E558A compared to WT (Fig. 3G-J) indicates an impaired function of POLD3 at telomeres, but this needs to be addressed. Why does MIDAS not increase with increased POLD3 at telomeres after expression of the E558A PARP2 mutant?

We have addressed this comment with comment #13 above.

17. The TRF of PARP1 and/or PARP2 KO over long-term passage should be moved to the main Figure.

We thank the reviewer for this suggestion. However, we have decided to keep this piece of data in the supplemental figure (Supplementary fig. 2b and 2c). Indeed, for better readability of the main figure panels, we believe it is essential to display the telomere dysfunction data in the main figure as they are key starting points for the rest of the study.

18. Lines 312-323 referring to Figure 5A-E: The images, especially in 5D, show PLA staining well outside of the nucleus. Are these also POLD3/PARP2 PLA? What is the explanation for this?

PLA staining outside the nucleus can occur for multiple reasons. The PLA experimental process can be itself responsible for unspecific signal, as described in the article by Hegazy et al. (Hegazy et al., 2020. *Curr Protoc Cell Biol*), either through the formation of crystal during components incubation, insufficient blocking, or non-specific antibody binding. Knowing these possible issues, we assure the reviewer that we follow a very rigorous procedure to perform our PLA, which also includes the systematic inclusion of antibody controls (illustrated in our supplemental figure). We have now acquired extensive experience in conducting PLA staining and we have noticed that the detection of over-expressed tagged proteins may sometimes lead to the formation of foci that are not confined to the nucleus. Nonetheless, our quantifications are confined to the nuclei that we select as Regions Of Interest using the NIS Elements software prior to applying the object count algorithm (See details in the Methods section). This ensures that our quantifications only concern the PLA foci formed between our 2 targets.

19. Line 337: the text does not explain why BLM depletion would increase POLD recruitment.

Previous work has demonstrated that BIR depends on polD3 (Minocherhomji, S. et al.2015; Yang et al., 2020). Moreover, the de Lange lab has demonstrated that depletion of helicase BLM triggers replication stress which triggers BIR at telomeres during which PolD3 is recruited (Yang et al., 2020). We have edited our manuscript to mention this point.

Reviewer #2 (Remarks to the Author):

1. In this report, the authors suggest that PARP2 plays a unique role during break-induced replication at telomeres that is dependent on the presence and activity of PARP2 by targeting PolD3. However, the study ends with analysis of DNA synthesis during BIR but is not designed to be telomere specific, whereas the first 4 figures are specific to telomeric DNA damage.

In this article we use 2 ways to induce replication stress for which telomeres are particularly sensitive because of their high guanine base content within their lagging DNA strand: oxidative lesions (8oxoG in particular) and G-quadruplex formation through BLM depletion. 8oxoG was previously shown to induce replication stress at telomeres (Fouquerel et al.,). Similarly, BLM was shown to induce an increase of G4 at telomeres and subsequent telomere fragility due to replication stress (Zimmermann et al., 2014). Its depletion was then used by the De Lange lab as a mean to study the mechanisms of fragile telomere formation in a follow-up study (Yang et al., 2020). We confirmed the increase of fragile telomeres upon both chronic 8oxoG induction using our FAP tool and upon BLM depletion in our Figure 2, panels E and G. We agree with the reviewer that BLM depletion may also increase G4 structures in other G-rich genomic regions. Therefore, our PLA experiments illustrated now Figures 7b-c may involve non-telomeric events. Thus, we took advantage of the FAP-mCER-TRF1 construct expressed in our cells to quantify the number of PLA foci that colocalized with the mCer signal exclusively and similarly observe an increase of RFP/FLAG PLA foci in both cell lines upon BLM depletion (New Supplementary fig. 7b and 7c). This is an interesting data that shows that PARP2 and POLD3 interaction is not restricted to telomeres upon replication stress.

Other concerns include:

2. Page 5, Results section: Authors have not documented the APE1 inhibitor used.

We apologize for omitting to specify the APE1 inhibitor used. We have used APE1 inhibitor III (Sigma cat#262017). We have added this information in the text and the Methods section.

The explanation for the APE1i effect is stated as “This effect can be attributed to the ability of PARP1 to compete with APE1 for the binding of AP site” – This statement should be clarified.

Several studies from the Lavrik and the Wilson’s laboratories have demonstrated that PARP1 can bind AP sites and BER can be completed even in absence of APE1 (Kutuzov 2011; Kutuzov 2020; Prasad 2015; Moor et al., 2020). Our data showing that APE1 inhibition does not prevent PARP1 association at the site of lesions at telomeres can therefore be interpreted as the contribution of PARP1 in the recognition of the AP site left unprocessed upon APE1 inhibition. We have edited the text to clarify our previous statement.

3. Page 5, Results section: Not clear on the statement “Single-cell clone populations ...”. Did the authors select single-cell clones or populations?

We apologize for not using the right terms.

For all our knock-outs and complemented cells, we have systematically grown stable cell lines from single cell. We have removed the term “population”.

4. Page 6: Fig 3A does not show the BLM siRNA immunoblot results as indicated. Should be Fig 1K.

We apologize for omitting to update our figures panel numbering. We have now edited.

5. Page 8: What is meant by the phrase “the PARP2 catalytic mutant in fusion with a flag tag”?

We believe the reviewer is referring to the sentence “We also complemented PARP1/2ko cells with the PARP2 catalytic mutant in fusion with a flag tag and (E558A) to evaluate the role of PARP2 catalytic activity”. To test the involvement of PARP2 in the occurrence of telomere fragility we have complemented the PARP1/2KO cells with either wild type PARP2 or its catalytic mutant (E558A) unable to perform Poly(ADP-ribosylation). Both proteins are expressed in fusion with a flag tag. We have rephrased.

6. Throughout: “Fig.s” should be changed to “Figs.”

We have edited.

7. Fig 3A: Authors should also include a direct comparison of the WT cell line to ensure that the level of expression of the transgene is similar to the endogenous level of expression for PARP2.

We have now run a new protein gel and western blot to include the HeLaFAP cell extract.

The data show that the expression of the transgenes is higher than in the HeLaFAP cells (new Figure 3b). However, to address additional comments from the reviewers (R1, comment #6; R2 comment #11 and R3, comment #3) we have performed experiments with additional clones for each cell line that display different telomere sizes (R1, comment #6), and lower transgene expression level and PARP2-dependent PARylation activity (R3, comment #3) (see western blots for protein expression and PARylation activity on new Supplementary fig. 3g and 3h).

Using these additional clones, we have conducted the following experiments after exposure to chronic oxidative stress:

- Metaphase spread followed by FISH to quantify the number of fragile telomeres and signal-free ends after chronic oxidative stress (new Supplementary fig. 3i and 3j)
- Metaphase spreads after EdU pulse treatment to quantify BIR-dependent conservative DNA synthesis (new Supplementary fig. 4d and 4e).
- Metaphase spreads followed by CO-FISH staining to determine whether BIR-dependent DNA synthesis is responsible for the fragile phenotype (new Supplementary fig. 5h and 5i).
- PLA for POLD3:TRF2 (New Supplementary figure 7b)
- Recruitment of Rad51 at telomeres following chronic oxidative stress (new Supplementary fig. 6e).

In summary, regardless of the size of telomeres, PARP2 re-expression and activity levels, we were able to draw the same conclusions than with the clones originally used.

We invite the reviewer to refer to the new manuscript for more details about the new data collected.

8. Page 9: The phrase "... MiDAS arose dependently of PARP2 activity (Fig. 3)" should be corrected to read as "... MiDAS arose dependent on PARP2 activity (Fig. 3)".

Thank you. We have edited accordingly.

9. Fig 5F: This data shows that PARP2 can modify PolD3 when isolated. Authors should also show that PolD3 is adp-ribosylated when cell are treated with ROS or the FAP.

To address this comment, we have performed a PLA assay using the PolD3 and a mono/poly-ADPr antibody in BLM-depleted cells.

We observed the formation of PLA foci in all our cell lines after treatment except in PARP1/2KO and in cells re-expressing the PARP2 E558A mutant (new Figure 7g and 7h) indicating that both PARP1 and PARP2 seem capable of targeting PolD3. Our data are in line with a recently published article from the Lakin lab (Richards et al., 2023) that demonstrated PolD3 ADPr in U2OS cells undergoing replication stress triggered by HU treatment. However, this study did not address the individual contribution of each enzyme.

Here, while PARP1ko (clone 6.10) cells did not seem to undergo a reduction in the number of PLA foci, PARP2KO (clone 22) cells displayed a significant decrease. These data suggest a more significant contribution of PARP2 in the ADP-ribosylation of PolD3 in our treatment conditions.

10. Top of page 5: The authors indicate "... antibody revealed that both PARP1 and PARP2 are efficiently recruited to telomeres immediately after acute dye and light exposure (Figs 1B and 1C)." but also should list Fig 1D.

We thank the reviewer for this thorough review of our article, and we apologize for not updating our figure panel numbering. We have now corrected the issue.

11. Fig 1E and throughout: It is expected that at least two KO clones are to be evaluated.

Please see response on comment #7 above.

12. Bottom of page 5: The authors indicate "...fragility compare to other cell lines (Fig. S1F),..." but this should be corrected to read "... fragility compared to other cell lines (Fig. S1F), ...".

We have edited accordingly.

13. Fig 1G: Is the decreased growth in the PARP1/PARP2-dKO additive as compared to the PARP1-KO and PARP2-KO or synergistic?

We have extended our statistical analysis of our data (New Figure 2b). As previously noted, PARP1 depletion impacts cell growth of treated cells but not PARP2 depletion. However, depletion of both enzymes has a significantly greater impact on cell growth than upon their

individual depletion. We explain this synergy by a role of both enzymes in promoting the response to replication stress via distinct pathways. PARP1 was found to promote repair of collapsed replication forks upon BLM depletion through the alt-EJ pathway (Young et al., 2021) while our data demonstrate that PARP2 promotes the BIR pathway in these same conditions as well as upon replication stress induced by chronic oxidative stress. We have discussed these points in the Discussion.

14. Fig S1I: The table should be labeled.

This table is now found in the new Supplementary fig. 2c.

15. Fig S1J: Why were the individual KOs not evaluated. This should be included.

The reason why the individual KOs were not evaluated is a practical one. TESLA assay requires the use of specific restriction enzymes provided by NEB. The enzyme CviAll was discontinued rendering difficult to conduct the assay with all samples. TESLA is mainly used on cell lines harboring particularly short telomeres. Since the double ko had the shortest telomeres to begin with, we prioritized these cells. To prevent any confusion from the readership and because this assay only complements our MTL assay (Supplementary fig. 2b and 2c), we decided to remove it from our data set.

16. Fig 3 (bottom of page 6): I note that this figure is mentioned after Figure 1 is presented but before Figure 2 is presented in the Results section. However, it appears the authors meant to refer to Fig 1L (not Fig 3) and it is also noted that Fig 1K is not mentioned in the text.

Our apologies for omitting to update our figure numbering. It is now corrected.

17. Fig 2I: Why were the other two cell lines (PARP2-KO and PARP1/2-KO) not evaluated for fragile telomere per metaphase? This should be included.

These data are available for viewing in Supplementary figure 5.

18. The authors should also evaluate PARP1/2 inhibitors vs PARP1 selective inhibitors.

This is an interesting suggestion that we had already considered and tested.

In this experiment we used the PARP1 and PARP2 inhibitor Veliparib and compared it to the recently developed PARP1-specific inhibitor AZD5303. We triggered replication stress using BLM siRNA in HeLaFAP cells that express both PARP1 and PARP2 and in PARP1KO cells and recorded the number of fragile telomeres.

In line with our previous observations, siBLM triggered an increase of fragile telomeres in both HeLaFAP and PARP1KO cells (see graph below). Moreover, consistent with a decrease of telomere fragility when depleting PARP2 in PARP1KO cells, treatment of PARP1KO cells with Veliparib led to a rescue in the number of fragile telomeres but not upon treatment with AZD5303 that does not inhibit PARP2. These data confirm a role for PARP2 activity in the mechanisms driving telomere fragility. However, we also noted a small but significant increase of fragile telomeres in HeLaFAP cells treated with both Veliparib and AZD5303 prior to siBLM treatment. An increase of telomere fragility following PARP inhibition using Olaparib, another PARP1 and PARP2 inhibitor, has also previously been reported by another lab (Maresca et al., Communications Biology, 2023).

Our explanation of this seemingly contradictory data pertains to the impact of PARP inhibitors on the allostery of the enzyme. Indeed, several PARP inhibitors have the potential to trap PARP1 and PARP2 on DNA based on their potency for inhibiting their catalytic auto-modification that is required for their rapid release from a DNA break (Murai et al. Cancer research 2012; Zandarashvili et al., 2020. Science). PARP trapping has been extensively described in the previous literature to trigger replication stress. It is therefore possible that telomere fragility, in our

conditions, is triggered by the trapping of the enzymes themselves via a mechanism that is independent of BIR.

Because the goal of our study is to demonstrate a role of PARP2 in driving the BIR mechanisms, we believe that although very interesting, demonstrating our hypothesis of PARP trapping-mediated telomere fragility would require more investigation that is beyond the scope of our study. Thus, we have decided to share our data with the reviewer but made the decision to not include it in our data set.

19. Figs 3B-J: Does re-expression of PARP1 give the same effect as re-expression of PARP2?

This is a very interesting question. We have complemented our PARP1/2Ko cells with a plasmid allowing the expression of PARP1 in fusion with a flag tag (new Figure 3a) and scored the number of fragile telomeres upon BLM depletion. We found that PARP1 re-expression rescued telomere loss as much as following PARP2 re-expression. However, while PARP2-FLAG restored telomere fragility, PARP1 re-expression did not promote this phenotype. Our data indicate that only PARP2 orchestrates the mechanisms leading to telomere fragility (New Figures 3d and 3e).

20. Fig 4: The cells labeled as “FLAG PARP2” and E558A”, are these the PARP1/2-KO complemented cells? If so this should be clarified.

We apologize for the confusion. Indeed, PARP2-FLAG and its catalytically dead mutant have been re-expressed in PARP1/2KO cells. We have clarified it in the paragraph related to the new figure 3.

21. Bottom of page 9: The statement “... demonstrated that telomere fragility and MiDAS arose dependently of PARP2 activity...” should be corrected to read “... demonstrated that telomere fragility and MiDAS is dependent on PARP2 activity ...”.

This sentence was cut out during the editing of the text.

22. Fig 5: Authors should show that PoID3 is PARylated in cells under the conditions used herein for replication stress.

See response to comment #9 above.

23. Page 11: What is meant by the phrase “requires a direct interaction with PoID3 and its PARylation”?

Here, we meant that the DNA end resection of BIR requires (1) an interaction between PARP2 and PoID3 and (2) PoID3 ADPr-ylation. This sentence was removed following editing of the manuscript.

24. The authors use the designation 'ko' for knock-out but the term normally used is 'KO'. Also, this should be defined.

We have corrected to KO. Ko was defined as “knock out” in our text pertaining to figure 1.

25. Throughout: Some English language editing is suggested.

We have edited our article to correct the grammatical mistakes and our native English speaker authors have performed a review to edit phrasings which may not have been idiomatic.

Reviewer #3 (Remarks to the Author):

This manuscript elucidates the role of PARP2 in fostering telomere fragility and averting telomere loss during replication stress. It proposes that PARP2 regulates telomere fragility through two stages of the break-induced replication (BIR) pathway: DNA end resection and DNA synthesis. The findings indicate that while the DNA end resection phase of BIR is not contingent on PARP2 catalytic activity, PARP2 aids BIR-dependent mitotic DNA synthesis through the PARylation of the Pol delta subunit POLD3. The paper contends that PARP2 is central to managing replication stress at telomeres and that it can induce telomere fragility through the BIR pathway. This study is compelling given its exploration of further roles of PARP2 in response to replication stress. However, the representation of dysfunctional phenotypes indicates that PARP2's role in mediating replication stress might be negligible, lessening the impact of the study. Additionally, the evidence for PARP2's role in prompting telomere fragility through PARylation-independent and -dependent steps of the BIR pathway lacks potency.

Major Points:

1. Figures 1B through 1D imply greater efficiency in the recruitment of PARP1 to telomeres in comparison to PARP2 during replication stress.

The goal of this initial experiment is to assess whether both PARP1 and PARP2 are first responders to acute oxidative lesion induction at telomeres. Replication stress at this time point (right after treatment) is minimal. Our data indicate that both PARP enzymes are efficiently recruited to the telomeres right after dye and light treatment indicating that both are involved in the first response of the cells to 8oxoG induction. PARP1 is much more abundant than PARP2 which can explain the lower number of PLA foci formed between PARP2 and TRF2.

Furthermore, chronic exposure treatment decelerates the expansion of PARP1-KO cells, while the proliferation rate of PARP2-KO cells remains steady (Fig. 1G). These findings strengthen the premise that PARP1 is the primary factor in managing replication stress, a theory reinforced by substantial previous studies. Conversely, PARP2 might have a secondary role in replication stress regulation.

The reviewer is right that PARP1's role in the response to replication stress has been extensively described in the prior literature. However, the prior literature also reports a role for PARP2 in this process and more evidence has emerged in the past 10 years. For instance, the Helleday lab reported in 2009 a role for both PARP1 and PARP2 in the response of cells to HU treatment (Bryant et al., 2009). Moreover, several studies have shown that PARP2 is activated by replication intermediates (Langelier et al., 2014; Obaji et al., 2016; Sukhanova et al., 2016; Kutuzov et al., 2013). A study from the Pennaneach lab has also highlighted a role for PARP2 in the DNA end resection mechanisms upon DSB induction that could be also formed upon replication fork collapse (Fouquin et al., 2017). Furthermore, a very recent study from the Lakin lab has reported a role for PARP1/2 in BIR upon HU treatment, although without singling out the role of each enzyme in this process (2023). Finally, work from the Yelamos lab has demonstrated that loss of

PARP2 triggers replication stress in erythroblasts that are highly proliferating cells (Farres et al., 2015) and that PARP2 limits c-Myc-driven B-cell lymphoma expansion by preventing c-Myc-mediated replication stress (Galindo-Campos et al., 2022). Collectively, these premises point towards a significant role for PARP2 in the cellular response to replication stress. These studies are mentioned in the introduction. Yet, we agree that PARP2's contribution in particular has been poorly explored up to now, not because it is not implicated in the replication stress response but most probably because PARP1 is the most abundant and active DNA-dependent ARTs. This is the gap that our study attempts to narrow down. Deciphering the mechanisms that PARP2 orchestrate in this response will surely encourage the development of new therapeutic approaches differentially targeting DNA-dependent ART enzymes, especially in cancer cells harboring high levels of replication stress. We have summarized these important points in our introduction.

We attribute the fact that PARP2KO cell's growth is less impacted by chronic oxidative stress to the presence of PARP1 that efficiently repair oxidative DNA lesions, thereby preventing the subsequent replication stress. To illustrate this, we have looked at the repair rate of oxidative lesions on denaturing southern blot (new Supplementary fig. 1f and 1g). Consistent with higher efficiency of PARP1 for the immediate repair of oxidized bases, our data showed an overall stronger induction of breaks in PARP1KO cells right after dye and light treatment than in PARP2KO cells.

2. Lines 160-161 in the manuscript conclude that findings suggest a role of PARP2-dependent PARylation upon exposure to a dye and light. However, no PAR signal is detected at telomeres post the treatment in PARP1-KO cells expressing PARP2, suggesting that PARP2 doesn't contribute to the major PARylation events during replication stress. Hence, the presented conclusions lack support from the results.

The goal of the assay the reviewer is referring to, was to assess whether both PARP1 and PARP2 are first responders to oxidative lesion induction at telomeres. Replication stress at this time point (right after treatment) is minimal. PARP1 is responsible for 90% of PARylation immediately upon damage. Thus, we were expecting to not be able to detect a significant signal coming from PARP2 activity at telomeres that represent only 0.05% of the genome. However, H₂O₂ treatment allowed us to detect PARP2 activity by western blot in PARP1KO cells (Supplementary fig. 1c). Moreover, our data also demonstrate that PARP2 responds to replication stress by orchestrating BIR during which it controls end resection independently of its catalytic activity (see new Figure 4) indicating that ADP-ribosylation is not a pre-requisite for PARP2 to hold a role in promoting a pathway.

In fact, several years ago, PARP1 has been reported to be a transcriptional regulator for which its PARylation activity is also not always required (Kraus 2008) such as during the co-activation of the NF- κ B factor upon an inflammatory stress (mostly work from the Hottiger lab: Hassa et al., 2001; Hassa et al., 2003; Hassa et al., 2005).

Moreover, the authors propose that PARP2 advances BIR-dependent mitotic DNA synthesis through PARylation of the Pol delta subunit POLD3, suggesting an expected PARylation signal in PARP1-KO cells expressing PARP2 under replication stress. This inference contradicts the results demonstrated in Figures S1D and S1E.

As mentioned above, these figures illustrate data obtained immediately upon acute exposure during which replication stress is minimal. Our experiments performed upon chronic oxidative stress or BLM depletion do trigger replication stress and therefore do not contradict these results. Following a comment from reviewer 2 (comment #9) we have now performed PLA experiments to assess of the ADP-ribosylation of PolD3 in cells upon induction of replication stress. Formation of PLA foci between ADPr and PolD3 antibody was greater in cells lacking PARP1 or in PARP1/2KO cells complemented with PARP2-FLAG than in cells lacking PARP2. This is consistent with a role of PARP2 activity in the BIR process upon replication stress.

3. The authors illustrate how PARP2 governs telomere fragility using PARP1/2-KO cells complemented with either wild or mutant type PARP2. However, the PARP2-WT or Mutant proteins in these cells are overexpressed, as evinced by Figure S3A. A similar magnitude of PARylation level in the re-expressed PARP2-WT-PARP1/2-KO cells as that in the PARP1-KO cells expressing endogenous PARP2 is presumable. However, the PARylation level in the re-expressed PARP2-WT-PARP1/2-KO cells is excessively high, even comparable to parental HeLa cells. Hence, this indicates an overexpression of PARP2 in the re-expressed PARP2-WT-PARP1/2-KO cells. Since most known functions of PARP2 duplicate those of PARP1, the overexpressed PARP2 could potentially imitate the functions of PARP1 rather than performing as an authentic PARP2 entity. Hence, the evidence with FLAG-PARP2 complemented in PARP1/2KO cells appears less persuasive. Before further consideration, the authors should select cell clones exhibiting comparable endogenous PARP2 expression and reiterate all the experiments.

The reviewer raises a very good point. To answer this concern, we have performed experiments with additional clones for each cell line that display different telomere sizes (R1, comment #6), and lower transgene expression level and PARP2-dependent PARylation activity (R3, comment #3) (see western blots for protein expression and PARylation activity on new Supplementary fig. 3g and 3h).

Using these additional clones, we have conducted the following experiments after exposure to chronic oxidative stress:

- Metaphase spread followed by FISH to quantify the number of fragile telomeres and signal-free ends after chronic oxidative stress (new Supplementary fig. 3i and 3j)
- Metaphase spreads after EdU pulse treatment to quantify BIR-dependent conservative DNA synthesis (new Supplementary fig. 4d and 4e).
- Metaphase spreads followed by CO-FISH staining to determine whether BIR-dependent DNA synthesis is responsible for the fragile phenotype (new Supplementary fig. 5h and 5i).
- PLA for POLD3:TRF2 (New Supplementary figure 7b)
- Recruitment of Rad51 at telomeres following chronic oxidative stress (new Supplementary fig. 6e).

In summary, regardless of the size of telomeres, PARP2 re-expression and activity levels, we were able to draw the same conclusions as with the clones originally used. We invite the reviewer to refer to the new manuscript for more details about the new data collected.

Finally, to further make sure that our results do not only result from a strong PARylation activity regardless of the enzyme responsible for it, we have also complemented the PARP1/2KO cells with PARP1, expressed in fusion with a flag tag (New Figure 3a) and scored the number of fragile telomeres upon BLM depletion. Consistent with a lack of a role for PARP1 and its activity in the occurrence of telomere fragility, the number of fragile telomeres did not increase unlike what we observed in cells complemented with PARP2 (New Figure 3e and 3g). These new data strengthen our conclusion that PARP2 alone orchestrates BIR-mediated telomere fragility. We invite the reviewer to refer to the new manuscript for more details about the new data collected.

4. The Western blot in Figure 5F fails to convincingly signify POLD3 as a likely PARP2 substrate. For enhanced evaluation, the Western blots should include molecular weight markers. Moreover, the observed PAR signal might stem from PARylated-PARP2 rather than PARylated-POLD3. The authors should conduct mass spectrometry to verify if POLD3 is indeed a potential PARP2 substrate. Furthermore, an investigation of whether POLD3 is a substrate of PARP1 rather than PARP2 should be undertaken.

Following the reviewer comment and comments from other reviewers (see Reviewer 1, comments 13 through 16, and reviewer 2, comment #9), we have performed several additional experiments to bring strength to our conclusions regarding PolD3 as a PARP2 target.

First, we agree with the reviewer that the signal obtained in our initial PARP2 hetero-modification assay using RFP-POLD3 pulled down from cells “might stem from PARylated PARP2”. In this assay we used recombinant purified PARP2 fused to GST (BPS Bioscience). Because GST and PARP2 molecular weights are respectively 26 and 66 KDa and RFP and POLD3 molecular weights are 27 and 66KDa, both proteins in the assay migrate at the same level. Thus, the signal could come from both auto-PARylated PARP2-GST and hetero-modified RFP-POLD3.

To go around this issue, we have now conducted a new PARP2 hetero-modification assay using Polymerase Delta recombinant protein complex that we obtained from our collaborators instead of pulled-down proteins. The purified Polymerase Delta (Pol δ) complex comprises the subunits p129 (POLD1), p66 (POLD3) and p50 (POLD2) (New Supplementary fig. 7f). We incubated PARP2-GST with NAD⁺ and activated DNA which led to the formation of smeary ADPr signal. Upon addition of Pol δ we observed additional smears and bands that indicated that not only POLD3 is ADP-rybosylated by PARP2 but also the other 2 subunits. Additional controls included the omission of one of the 4 factors used in the assay (NAD⁺, PARP2, or Pol δ or activated DNA) showing the specificity of our signals (New Figure 7f).

To complement this assay and show ADP-ribosylation of POLD3 in cells, we have also performed a PLA assay using ADPr antibody and POLD3 antibody in all our cell lines treated with BLM siRNA (New Figures 7g and 7h). We observed the formation of PLA foci in all our cell lines after treatment except in PARP1/2KO and in cells re-expressing the PARP2 E558A mutant (New Figures 7g and 7h) indicating that both PARP1 and PARP2 seem capable of targeting PolD3. Our data are in line with a recent study from the Lakin lab that was published after our initial submission (Richards et al., 2023) that demonstrated PolD3 ADPr in U2OS cells undergoing replication stress triggered by HU treatment. An earlier system-wide study from the Nielsen’s lab had also identified POLD3 as a target for Serine ADP-ribosylation (Larsen et al., 2018). However, these studies did not address the individual contribution of each enzyme.

Here, while PARP1ko (clone 6.10) cells did not seem to undergo a reduction in the number of PLA foci, PARP2KO (clone 22) cells displayed a significant decrease. These data suggest a more significant contribution of PARP2 in the ADP-ribosylation of PolD3 in our treatment conditions but do not exclude the contribution of PARP1 in cells. We added this point of discussions in our edited manuscript. Finally, while mass spectrometry would reveal that PARP2 and POLD3 function in the same pathway it would not confirm that POLD3 is a direct substrate for PARP2. However, Mass spectrometry is included in our future experimental plans to identify the protein network put in place under conditions of replication stress at telomeres.

Minor Points:

1. To indicate their precise position, the Western blots should incorporate dashed lines at the positions of molecular weight markers.

We have now added the molecular weights on all our western blot images.

2. Proper presentation of the references cited in lines 129 and 148-149 is necessary.

We apologize for omitting to deleting some of the references after formatting. We have corrected it.

3. The image quality of Figure 1A, especially the text, needs improvement.

We have increased the font size and the resolution of the schematic.

4. The study should provide p-values for Figures S3C and S3D.

We have now added the p-values.

5. The text contains numerous mislabeling, for instance, there is a reference to "Fig. 7" in

line 379. However, “Fig. 7” doesn't exist in the manuscript.
We apologize for these mistakes. We have edited.

REVIEWERS' COMMENTS

Reviewer #1 (Remarks to the Author):

The authors have addressed the majority of my comments, and I am satisfied with the modifications to the manuscript. In particular, I appreciate the further examination of the telomere length variable and the overall strengthening of the conclusions.

The manuscript does require some typographical editing, particularly to the additional/amended sections.

Reviewer #2 (Remarks to the Author):

Overall, the authors have addressed the major and minor concerns raised by this reviewer.

Reviewer #3 (Remarks to the Author):

The authors have addressed my questions, and I recommend the manuscript for publication.

Reviewer #1 (Remarks to the Author):

The authors have addressed the majority of my comments, and I am satisfied with the modifications to the manuscript. In particular, I appreciate the further examination of the telomere length variable and the overall strengthening of the conclusions. The manuscript does require some typographical editing, particularly to the additional/amended sections.

We thank the reviewer for their thorough examination of our resubmitted article and for their help in improving it.

We have edited the new manuscript.

Reviewer #2 (Remarks to the Author):

Overall, the authors have addressed the major and minor concerns raised by this reviewer.

Reviewer #3 (Remarks to the Author):

The authors have addressed my questions, and I recommend the manuscript for publication.